# Understanding How Chemical Pollutants Arise and Evolve in the Brewing Supply Chain: A Scoping Review

**DOI:** 10.3390/foods13111709

**Published:** 2024-05-29

**Authors:** Gabriel Pérez-Lucas, Ginés Navarro, Simón Navarro

**Affiliations:** Department of Agricultural Chemistry, Geology and Pedology, School of Chemistry, University of Murcia, Campus Universitario de Espinardo, E-30100 Murcia, Spain; gpl2@um.es (G.P.-L.); gnavarro@um.es (G.N.)

**Keywords:** chemical pollutants, beer, brewing, health risk, raw materials

## Abstract

In this study, a critical review was carried out using the Web of Science^TM^ Core Collection database to analyse the scientific literature published to date to identify lines of research and future perspectives on the presence of chemical pollutants in beer brewing. Beer is one of the world’s most popular drinks and the most consumed alcoholic beverage. However, a widespread challenge with potential implications for human and animal health is the presence of physical, chemical, and/or microbiological contaminants in beer. Biogenic amines, heavy metals, mycotoxins, nitrosamines, pesticides, acrylamide, phthalates, bisphenols, microplastics, and, to a lesser extent, hydrocarbons (aliphatic chlorinated and polycyclic aromatic), carbonyls, furan-derivatives, polychlorinated biphenyls, and trihalomethanes are the main chemical pollutants found during the beer brewing process. Pollution sources include raw materials, technological process steps, the brewery environment, and packaging materials. Different chemical pollutants have been found during the beer brewing process, from barley to beer. Brewing steps such as steeping, kilning, mashing, boiling, fermentation, and clarification are critical in reducing the levels of many of these pollutants. As a result, their residual levels are usually below the maximum levels allowed by international regulations. Therefore, this work was aimed at assessing how chemical pollutants appear and evolve in the brewing process, according to research developed in the last few decades.

## 1. Introduction

After water and tea, beer is the most consumed beverage overall and has been associated with human culture for millennia. The global consumption of beer in 2020 was 177.5 million kL, decreasing by about 12.8 million kL due to COVID-19 diffusion. By 2021, the global beer market (around 186 million kL) had recovered almost half of the volume lost at the peak of the pandemic in 2020. In line with a market research report available by Zion Market Research [1], the analysis of the global beer market size and share revenue is valued at up to about USD 750 billion in 2021 and is projected to grow to around USD 815 billion by 2028, at a CAGR of around 6% between 2022 and 2028.

Beer is an alcoholic beverage produced by the saccharification of starch and the fermentation of the resulting sugar. The starch and saccharifying enzymes are provided by malted cereal grains, most commonly malted barley and malted wheat. Barley, hops, water, and yeast are the primary ingredients used in the production of beer [2]. Barley is the most commonly used grain for malting. However, malt can also be made from wheat, rye, and other grains. The process by which beer is made is called brewing or beer-making. Beer is also flavoured with hops, which add bitterness and act as a natural preservative. Other flavourings, such as herbs or fruits, may also be added. 

There are three stages in the malting process: steeping (to promote the progress of hydrolytic enzymes), germination (to maximise the amount of extractable matter produced by the enzymes), and kiln drying (to prevent germination and stabilise the malt by reducing the moisture content). In general, the commercial beer production process consists of the following steps, as summarised in Figure 1 [3]: (i) the dirt from the malt is removed during the air conveying system into the crushing process; (ii) the starch from the malt from the above process is put into the mashing cup and water is added, and the temperature of the starch solution is raised by the heat from the boiler until it reaches boiling temperature; (iii) after the heating process, the output from this is transferred to the mash tun by keeping it at an exact temperature to develop saccharification; (iv) it is then transferred to the mash filter to remove the spent grains from the solution, and the output of this process is called “sweet wort”; (v) after 1 h of boiling, the clear hot wort (brewer wort) is passed through a cooling system and clean air, yeasts (*Saccharomyces carlsbergbensis* or *uvarum*) are added, and the mixture is kept in storage tanks; (vi) after 2–3 weeks, the carbohydrates in wort are mainly transformed into C_2_H_6_O and CO_2_ by the yeasts in the fermentation tank and the storage tank; (vii) it is then transferred to beer filter tanks to remove yeast and other sediments to obtain clear beers; and (viii) the clear beers are packaged in cans, bottles, and kegs to become the final beer products.

Nowadays, the benefits of reasonable beer consumption for human health are increasingly emphasised due to the presence of positive properties, such as significant amounts of minerals, antioxidants, vitamins, and other healthy substances, as well as a low sugar content [4]. Many studies show that moderate beer’s drinkers have lower death rates from all causes, but particularly from cardiovascular disease, than heavy drinkers and those who do not drink at all. However, many toxic substances can be found in the raw materials used to make beer, such as water, malt, or hops. Therefore, with the increase in demand for beer, there is a need for research and special attention to possible pollutants that could have an impact on consumer health. 

Microbial pollution of beer is not well documented but should not be ignored because of the important health implications [5]. The term coliform bacteria refers to the *Enterobacteriales* order, a group of bacteria that produce acid and gas by fermenting lactose (at 30–37 °C within 48 h). This group includes *Escherichia coli* and other species of the genus *Escherichia,* in addition to other genera such as *Enterobacter*, *Citrobacter*, *Klebsiella*, *Raoultella,* and some members of the genus *Serratia*. Some species of enterobacteria can reduce nitrates to nitrites, which then react with amines to form carcinogenic nitrosamines. However, although coliforms are present in wort, their sensitivity to ethanol and acidic pH (4.1–4.5) prevents their transfer to finished beer [6]. 

On the other hand, chemical pollution can result from the cultivation of barley and other cereals and hops, as well as during grain storage, processing, and packaging. Some pollutants can also come from the soil itself and from the water used, since this is the largest component of the beer (90%) [7]. The main chemical pollutants found throughout the brewing process are biogenic amines, heavy metals, mycotoxins, nitrosamines, phthalates, bisphenols, pesticides, acrylamide, and, more recently, microplastics and, to a lesser extent, hydrocarbons (aliphatic chlorinated and polycyclic aromatic), carbonyls, furan-derivatives, polychlorinated biphenyls, and trihalomethanes. All are summarised in Table 1, where their source of pollution and their toxicological effects are fully described. 

It is therefore essential that all breweries have a hazard analysis and critical control points (HACCP) programme in place—a preventative, methodological approach to the safety of beer that controls critical points and addresses risk through prevention. Consequently, an understanding of the presence of chemical pollutants in beer production is a fundamental part of the modern brewing industry [8]. Therefore, this work was aimed at evaluating how chemical pollutants appear and evolve in the brewing process, according to research developed in the last few decades.

## 2. Methodology

The data used for this review were obtained from the Science Citation Index Expanded (SCIE) of the Web of Science Core Collection (WoSCC) database, which is the oldest bibliographic data source in the world and the most comprehensive and widely used source for research evaluation and analysis [9]. The analysis included records resulting from a systematic search for documents using the following combination of search terms “brewing” OR “beer-making” AND “biogenic amines” OR “pesticid*” OR “mycotoxin*” OR “metal*” OR “polycyclic aromatic hydrocarbons” OR “nitrosamine*” OR “polychlorinated biphenyls” OR “furan*” OR “carbonyl*” OR “phthalate*” OR “bisphenol*” OR “microplastics” OR “aliphatic chlorinated hydrocarbons” OR “acrylamide” OR “trihalomethane*” in the subject (title, abstract, and indexing), full text, and cited references. Extensive standardisation was applied for this analysis to minimise the risk of bias. In addition, other relevant references to the structures, modes of action, or toxicological data of the pollutants studied have also been included in this review. Therefore, to illustrate the behaviour and fate of these chemical pollutants during beer brewing, some relevant studies are discussed in the following subsections.

## 3. Behaviour and Fate of the Most Important Chemical Pollutants in the Course of the Beer Production Process

As mentioned above, biogenic amines, heavy metals, mycotoxins, nitrosamines, phthalates, pesticides, acrylamide, bisphenols, and, more recently, microplastics and other minority compounds, such as hydrocarbons (aliphatic chlorinated and polycyclic aromatic), carbonyls, furan-derivatives, polychlorinated biphenyls, and trihalomethanes, are the main chemical pollutants found during the beer brewing process, including malting. All structures shown in Figure 2, Figure 3, Figure 4, Figure 5, Figure 6, Figure 7 and Figure 8 are according to the CompTox Chemicals Dashboard of the US EPA [10]. 

### 3.1. Biogenic Amines

Beer is a complex combination of amino acids, proteins, and BAs that evolve and change during brewing. Yeasts use amino acids to create peptides and as a source of energy during the brewing process. This results in the removal, formation, and metabolism of amino acids, often producing BAs. Several studies have shown that microbial stressors such as nutrient deprivation and low pH can increase the production of BAs in fermented foods, but little has been published regarding beer.

Biogenic amines (BAs) are basic nitrogenous substances containing one or more amine groups, formed mainly from amino acids (by decarboxylation) and aldehydes and ketones (by amination and transamination). All are low-molecular-weight organic bases and compounds of biological importance in plant, microbial, and animal cells [11]. They are synthesised by microbial, plant, and animal metabolisms. They are produced by enzymes in the raw material or by microbial decarboxylation of amino acids in foods and beverages, leading to high human exposure to their toxic effects [12]. 

Low concentrations of BAs in food can be tolerated due to the detoxification process by intestinal amine oxidases. However, if high levels of BAs are found, the amine oxidases do not work properly to detoxify them, and this can lead to serious health problems [13].

In terms of their chemical structure, BAs are divided into the following three groups: (i) aliphatic BAs (putrescine, PUT, and cadaverine, CAD), (ii) aromatic BAs (tyramine, TYR, and phenylethylamine, PHE), and (iii) heterocyclic BAs (histamine, HIS, and tryptamine, TRY). Another classification is based on the number of aminating groups, as follows: (i) monoamines (TYR and PHE), (ii) diamines (HIS, PUT, and CAD), and (iii) polyamines (spermidine, SPD, and spermine, SPM), as shown in Figure 2 [12].

Several parameters during the beer process, such as raw material hygiene, microbial composition, and fermentation status and duration, influence the amount of BAs in foods and beverages [14,15]. The critical parameters for controlling the increase in BA concentration in processed products include temperature, additives, packaging, irradiation, hydrostatic pressure, pasteurisation, smoking, and starter culture [16]. The BAs are traditionally analysed by high-performance liquid chromatography (HPLC) with UV detection and pre-column derivatization using dansyl chloride (DNS-Cl) or 2-napthyloxycarbonyl chloride (NOC-Cl) as reagents. More recently, ultra-high-performance liquid chromatography coupled to mass spectrometry (UHPLC-MS^2^) and ultra-high-performance liquid chromatography with triple quadrupole time-of-flight mass spectrometry (UHPLC-QqQ-TOF-MS) techniques, operating in multiple reaction monitoring (MRM) mode, have been proposed [17]. 

It is important to know which BAs are present in beer and whether a particular style of beer is likely to have high levels of these compounds due to health concerns. Identifying the specific amino acids present in wort prior to fermentation and understanding which ones are most readily used by the yeast is a challenging task. However, it enables brewers to adjust their brewing practices to promote better products and minimise residual concentrations of BAs [18].

Buňka et al. [19] analysed 114 samples of beer from 28 Czech breweries for BA concentrations. The amounts of HIS, PHE, and TRY were very low (<30 mg L^−1^). SPM and SPD were also present at low levels. However, the levels of TYR, PUT, and CAD reached remarkable values, especially in alcoholic beers. In almost 25% of these beers, the total amount of BAs tested exceeded the limit of 100 mg L^−1^, which is considered to be of toxicological significance, especially in alcoholic beverages. Nalazek-Rudnicka et al. [20] determined 17 BAs in commercial beer samples to monitor concentration changes in several BAs during fermentation. Concentrations of BAs were above the safety threshold for consumers in some of the samples analysed. During the fermentation phase of the homebrewed ale, the concentration of SPM in the wort increased until the end of the stormy fermentation. At the end of this step, the concentration dropped below the initial concentration and below the LOQ after fermentation.

Some researchers investigating different Chinese beers have found large variations in the levels of some Bas, such as HIS, TRY, TYR, PUT, and SPD. The results of this study showed that the amines tested varied greatly. Some amines, such as HIS, were reported to have a variation of 480% between the highest and lowest concentrations in the beer samples [21]. Even though the concentration of BAs was considered low, the variations found were surprisingly different from each other. The variation found between the beers tested was surprising, even though the concentration of BAs was low. Major brands such as Heineken, Corona, and Yanjing (pH = 4.0–4.1; alcohol content (%) = 3.7–4.7; and original gravity = 10.0–11.3 ^0^P) were all tested, which would be expected to have more consistent concentrations of BAs. The level of processing, type of ingredients, and fast turnaround times of these beers are likely to explain the overall low concentration, which varied from 0.96–4.62, ND-1.62, 3.47–6.35, 2.12–5.73, and ND-1.41 µg L^−1^ of HIS, TRY, TYR, PUT, and SPD, respectively.

BAs and how they are formed in beer are not currently well known. One of the most well-known factors influencing the production of BAs is pH. Two studies have shown that when bacteria generate BAs such as TYR, bacteria’s ability to survive in acidic environments is greatly enhanced. In tests conducted at pH levels of 1–7, strains that produced TYR had 50% more viable cells than those that did not [22,23]. Chemically, this can happen because BAs are basic compounds. Therefore, when an amino acid is decarboxylated, not only is an acidic compound removed from the microbe’s environment, but a base is also produced. The production of this base is found in a wide range of lactic acid bacteria, including the genera commonly used in sour beer brewing, such as *Lactobacillus* and *Pediococcus* [24]. For example, due to the combination of lactic acid bacteria, yeast, and high acidity, some beers may have higher levels of BAs than others. 

### 3.2. Heavy Metals

Some metals, like potassium, calcium, magnesium, and iron, have nutritive properties and, at very low concentrations, are indispensable for the proper functioning of the human body, and their regular consumption may therefore be beneficial to the body from the point of view of mineral replacement. On the other hand, other metals such as Zn, Al, or Ni have no biological function and can cause serious diseases. A group of metals with a high toxicity degree (causing systemic damage to organs even at low levels of exposure) includes As, Cr, Cd, Pb, and Hg (priority HMs). 

Heavy metals (HMs), such as Cd, Hg, Cu, Cr, Pb, Ni, etc., are a grouping of metals and metalloids that, at ppb levels, are relatively dense and toxic [25]. Both natural and anthropogenic sources release these metals into the environment, including industrial discharges, vehicle emissions, and mining. HMs are not biodegradable and tend to bioaccumulate in living organisms. In fact, most are known to be potential carcinogens.

In spite of the regularity with which the major food and drink agencies such as the European Commission (EC-EFSA), the World Health Organization (WHO), the United States Food and Drug Administration (USFDA), the China Food and Drug Administration (CFDA), and the Food Safety and Standards Authority of India (FSSAI) regulate HMs in consumer products and animal feed, there are still many uncertainties about their effects. The quantification of the determination of HMs in food, beverages, and other matrices is carried out using a variety of analytical techniques. Their estimation depends on the property of the metal and its concentration in the sample to be analysed. The most commonly used techniques to analyse HMs in food are flame atomic absorption spectroscopy (FAAS), flame emission spectroscopy (FES), UV/VIS spectroscopy, inductively coupled plasma (ICP) with optical emission spectrometry (OES), mass spectrometry (MS) and atomic emission spectrometry (AES), atomic fluorescence spectrometry (AFS), and X-ray fluorescence spectrometry (XRFS) [26].

Metals in beer come mainly from raw materials, which can be polluted by metal-containing agricultural agrochemicals (fertilisers and pesticides) or by ecosystem pollution [27]. Brewing equipment (tanks, pipes, containers, and filters) or containers used to transport or store the finished product, including kegs and aluminium cans, can also be a source of metals in beer [28]. During beer production, depending on the technological process and the chemical composition of the intermediate products, the concentration of metals can vary [29].

Some results have shown that the raw materials (water, malt, and hops) have low (µg L^−1^) heavy metal pollution (mainly Cr, Cu, and Zn). Only a negligible fraction of these toxic metals can be detected in beer because most of them are transferred to the residues (spent grains, hot trub, and yeast). For example, because much of the Cu remains in the spent hops, treating the hops with a high dose of a Cu-containing pesticide did not increase the copper content of the final beer. In addition, there was an increase in the concentration of Cd, Fe, As, and Zn during the filtration phase [30]. The maximum limits of metals are derived from the Joint FAO/WHO Expert Committee on Food Additives (JEFCA), and the maximum concentration of tin in canned beer, 100 mg kg^−1^ is given by the Commission Regulation (EC) 1881/2006 [31].

The amount of HMs in the raw materials and their capacity to dissolve during the brewing process determine the concentration of HMs in sweet and hopped wort [27,30]. Metals can be essential or toxic and can also have an impact on the brewing process and beer quality for flavour stability and haze [32]. Some authors have shown that Zn concentrations can increase when raw materials or intermediate products, such as wort concentrate, are stored in cans [33]. According to the study conducted by Eticha and Hymete [32], the mean concentrations of HMs in beer produced in Ethiopia were as follows: 1.4 µg L^−1^ (Cd), 37 µg L^−1^ (Cu), 6.0 µg L^−1^ (Pb), and 1520 µg L^−1^ (Zn). The risk assessment of the mean values showed no health risk related with these HMs when beer is consumed by citizens. Other results discuss how Cu and Pb can be adsorbed simultaneously by brewers’ waste yeast because the amount adsorbed for one metal decreases as additional metals become available. However, the total capacity to bind HMs remains relatively constant. This suggests that ion exchange is one of the main mechanisms involved in the adsorption process [34]. Zufall and Tyrell [35] showed that Cu content is generally decreased during wort boiling and trub removal, and high Fe losses are observed during fermentation. Some authors [27] monitored the distribution of some HMs during beer production. Except for As, only a very small fraction of the other HMs entered the beer since the grain absorbed most of the metals (mainly Hg and Pb, at least in the case of As). A smaller proportion of metals, mainly Ni, As, and Se, were absorbed by the yeasts. Among the metals mentioned above, only As was transferred the most to the beer (about 2/3 of the amount added). On the other hand, the content of metals in the beer remained below the legal limits even after a large spiking of the brewing water.

### 3.3. Mycotoxins

Mycotoxins (MTs) are highly toxic compounds that are produced naturally by certain moulds (fungi). Many foods, such as cereals, nuts, dried fruits, or spices, grow moulds that can produce MTs. Mould growth can occur, often in warm, moist, and humid conditions, either before or after harvesting, storing, or in the food itself. Most of them are chemically stable and withstand food processing [36].

Although there are hundreds of MTs identified, fewer than ten attract the most attention because of their significant impact on human health and their presence in food. According to the WHO, the most frequently observed MTs of concern to human health and livestock include aflatoxins (AFs), ochratoxins (OTs), patulin (PAT), fumonisins (FMs), citrinin (CIT), zearalenone (ZEA), nivalenol (NIV), and deoxynivalenol (DON) (Figure 3). There are four compounds produced by *Aspergillus flavus* and *Aspergillus parasiticus* that belong to the AF class: AF B_1_, B_2_, G_1_, and G_2_ [37]. AF B_1_ is the major toxin produced, and it is considered the most toxic [38], while AF M_1_ is the main metabolite of AF B_1_ in humans and animals. 

The most common toxigenic fungi in Europe are *Aspergillus*, *Fusarium*, and *Penicillium* species. They produce AF B_1_, ochratoxin A (OTA), ZEA, FM B_1_, T-2 toxin, HT-2 toxin (both are type A trichothecene MTs, which are closely related epoxy sesquiterpenoids), and DON (vomitoxin, a mycotoxin of the type B trichothecenes epoxy sesquiterpenoids), which are of increasing concern for human health [39]. These MTs are continuously monitored in Europe, although the regulatory aspects still need to be harmonised at the European level. Mould infestation of crops, both before and after harvest, leads to MTs entering the food chain. Exposure to MTs can occur directly by consuming infected food or indirectly through animals fed contaminated feed, particularly milk as AF M_2_, a metabolite of AF B_2_. They can be transferred from cereals to cereal products such as malt and beer, exposing humans and animals to their effects [40]. This study summarises the latest knowledge on emerging MTs in barley, malt, and wort and their transfer to the final product, beer.

AFs, produced by certain moulds found in cereals such as wheat, barley, maize, sorghum, and/or rice, are among the most toxic MTs. OTA is produced mainly by *Aspergillus ochraceus* and *Penicillium verrucosum* during the storage of crops and is a common food-contaminating mycotoxin known to cause a range of toxic effects in animals. In 1993, OTA was catalogued as possibly carcinogenic to humans (group 2B) by the International Agency for Research on Cancer (IARC) [41]. PAT is a mycotoxin produced by a number of moulds, particularly *Aspergillus*, *Penicillium* and *Byssochlamys*. PAT can be found in various mouldy fruits, cereals, and other foods. FMs are produced by *Fusarium verticillioides* and *Fusarium proliferatum*. CIT is produced from several species of the genera *Aspergillus* (*A. ochraceus*), *Monascus*, and *Penicillium* (*P. verrucosum*), which frequently contaminate grain. Finally, ZEA, NIV, and DON are produced by *Fusarium graminearum* [42].

Before 1985, the Food and Agriculture Organisation (FAO) estimated the global contamination of food crops with MTs at 25%. To assess the basis for this, the relevant literature was reviewed by Eskola et al. [43], and data from around 500,000 analyses from EFSA and a large global survey for AFs, FMs, DON, T-2, and HT-2 toxins, ZEA, and OTA in cereals and nuts were examined. The current occurrence of MTs above EU and Codex Alimentarius limits seems to confirm the FAO estimate of 25%. However, the occurrence above detectable levels (up to 60–80%) is greatly underestimated by this figure. A combination of improvements in the sensitivity of analytical methods and the effects of climate change are likely to explain the high incidence. It is important not to overlook detectable levels. Humans are exposed to mixtures of mycotoxins through their diet, which can lead to a combination of adverse effects on their health.

Barley, malt, and beer are very complex matrices. Sample preparation and cleaning of the sample prior to the analysis of MTs are required. The most commonly used methods are liquid–liquid and solid–liquid extraction or immunoaffinity columns (IACs) [44]. The possibilities for detection of OTA and other MTs include high-performance liquid chromatography (HPLC), thin-layer chromatography (TLC), gas chromatography (GC), mass spectrometry (MS), capillary electrophoresis (CE), enzyme-linked immunoassay (ELISA) tests, and ultra-high-performance liquid chromatography (UHPLC) coupled to fluorescence detection (FLD) [45].

The pollution of barley and malt by fungi leads to a high economic burden due to the loss of malt yield, the loss of quality, and the costs associated with the presence of toxic fungal secondary metabolites (i.e., MTs). The fungal contamination of grain and malt, both in terms of quality and quantity, and the impact of fungal growth on the quality of malt and beer are influenced by a number of pathways and factors. Special emphasis will be placed on the role of the MTs and their fate during the brewing process, as well as on the latest research on beer gushing [46]. 

Several compounds of varying toxicity and abundance in cereals belong to the AF group. A maximum level for AFs has been recently actualised for cereals, cereal products, and dried vine fruit by Regulation (EC) 915/2023 [47]. The levels of AFs and OTAs in beer are under indirect control, as their levels in beer are a function of their presence in malt, for which a maximum level has been set. No maximum level for MTs in beer has been set by the European Commission (EC). Limit values as low as reasonably achievable are therefore recommended. It should be considered in the context of the review foreseen under Regulation (EC) No 123/2005 [48]. Technical knowledge and improvements in manufacturing and storing techniques do not prevent these forms from developing. Consequently, the presence of MTs in barley grains cannot be completely eliminated. Efforts to improve growing, harvesting, and storing practices to reduce mould growth should be encouraged. The most recent developments in the analytical toolbox, including chemical and molecular biological approaches, are of particular importance as they provide a powerful tool to prevent the ingress of fungi and their metabolites into the malt beer chain.

Many studies showed OTA content in barley ranging from 0.1 to 2.7 µg kg^−1^ [49], 0.01 to 0.5 µg kg^−1^ [50], and 0.5 to 12 µg kg [51]. The OTA level in malt varied from 0.1 to 0.9 µg kg [49] and 0.5 to 6.6 µg kg [51]. Some findings have advised that the level of *P. verrucosum* pollution is a good indicator of potential OTA pollution [52]. Lund and Frisvad [53] found that while there was no linear relationship between the two factors, grain samples with more than 7% *P. verrucosum* were indicative of OTA pollution. The main abiotic factors affecting the growth and OTA production of these spoilage fungi are temperature, water availability, and, if the grain is moist, gas composition [54]. Whether mould growth occurs and, if so, the relative development of the fungal community are largely determined by the interactions between these variables. A precise determination of the marginal conditions under which species such as *P. verrucosum* and *A. ochraceus* grow and produce OTA is important because it can be used to provide recommendations on the risk of contaminating grain through the food chain. 

Malachová et al. [55] evaluated the occurrence of *Fusarium* MTs in different barley cultivars. The variety and levels of target MTs changed during the experimental period. In 2005, DON was detected in most samples (mean 42 µg kg^−1^) together with HT-2 (mean 25 µg kg^−1^). In the following year, DON levels were lower (mean 13 µg kg^−1^), and contamination by HT-2 and T-2 was insignificant. In 2007, NIV (mean 45 µg kg^−1^) was detected in almost all samples, together with DON (mean 39 µg kg^−1^). The highest levels as well as occurrences of A-group trichothecenes, HT-2 (maximum level of 715 µg kg^−1^), and T-2 (maximum level of 320 µg kg^−1^) were detected in 2008, when the crop was possibly attacked by producers of these toxins due to the mild winter. Contrary to expectations, DON levels were higher in fungicide-treated barley than in untreated barley at harvest. Unfortunately, an active ingredient may reduce the growth of one type of pathogen and thus the development of another, resulting in increased production of MTs. This fact was examined in a study published by Simpson [56], where the use of azoxystrobin was effective in the control of the non-toxigenic fungus *Microchodium nivale* but ineffective in the control of *Fusarium head blight*. In contrast, fungicide may act as a stress factor for *F. culmorum* or *F. graminearum*, resulting in higher toxin production [57]. It should be noted that both *F. culmorum* and *F. graminearum* have been detected in fungicide-treated harvested barley [58]. HT-2 levels were reduced by solubilisation in the steeping water during the malting process. Changes in DON showed no general trend. The results were rather controversial, with both decreasing and increasing trichothecene occurring during malting. The effect of the malting process on the DON content of malt could not be generalised.

As a starting point for brewing, some breweries use pre-malted grain. This can affect the beer microbiota, as the grain may be stored for months before use [59]. Visible mould can be developed on malted grain after only one month of storage at high water activities (0.8–0.9), and it appears after three months of storage at slightly lower water activities (0.7), although malted grain can be stored for up to 12 months if the water activity is low (<0.5) [60]. According to the literature, appropriate storage conditions for cereal grains and malting and brewing by-products are undoubtedly important and should be provided, such as low water activity, temperature, and a shorter storage time [61]. 

Some of the *Fusarium* MTs present in infected barley, such as DON, may be lost during steeping. However, the *Fusarium* mould is still able to grow and produce MTs during germination and kilning. Consequently, unless further mould growth is also prevented, it may not be practical to detoxify the grain prior to the malting process. Methods are needed to reduce mould growth during malting. Physical (irradiation), chemical (ozonisation) and biological methods are available to inhibit mould growth in grain. Although biological control methods may be desirable, the effects of these inhibitors on malting and brewing quality need further investigation [62]. Gumus et al. [63] reported that in 26 out of 29 barley samples, the OTA concentration was found to be in the range of 0.53–12 μg kg^−1^. The mean value was 4.9 μg kg^−1^. They reported that 15% of barley samples contained less than 3.0 μg kg^−1^ OTA, 31% between 3 and 5 μg kg^−1^, and 54% more than 5.0 μg kg^−1^.

In brewing, rice grits, corn in the form of grits or syrup, unmalted barley, sorghum grits, or wheat starch may be used to provide fermentable carbohydrates for yeast. Some studies showed that AF B_1_, OTA, ZEA, DON, and FMs (B_1_ and B_2_) may be transmitted from the malted grain or from adjuncts to beer [64,65]. CIT fails to survive being crushed. In local beers brewed in Africa, high incidences and concentrations of AFs and ZEA were found. However, AFs were not detected in European beers. Except for one sample analysed by TLC only, ZEA and α- or β-ZEA (the likely metabolite) were not found in Canadian and European beers. OTA was rarely detected at a level higher than 1 ng mL^−1^ in beer. However, moderately high occurrences of trace levels have been reported using LC methods with a detection limit (LOD) of 0.05–0.1 ng mL^−1^. DON surviving the brewing process has been found at high levels in Canadian and European beers, with levels > 200 ng mL^−1^ recorded in several German beers. FMs (B_1_ and B_2_) occur to a reduced extent in beer. Krogh et al. [66] observed significant OTA losses (40–89%) in the grist during mashing, most likely due to proteolytic degradation. A further 16% was removed with the spent grains. In the course of fermentation, OTA losses varied from 2% to 69%, while the remaining OTA was transferred to the beer [67]. The OTA content of beer was first described in 1983 [68]. Several studies carried out worldwide since 1998 have reported OTA levels in beer varying from ND to 0.5 ng mL^−1^ [69], except for the 2002 study on South African beer [70], where the value of 2340 ng mL^−1^ was the highest stated for OTA in beer.

Chu et al. [71] found that the loss patterns of these MTs were similar when studying the stability of AF B_1_ and OTA when brewed. Both MTs were relatively stable at the boiling temperatures of the mash boiling step. However, they were more vulnerable to malting (protein hydrolysis), wort boiling, and final fermentation, and removal ranged from 12 to 27%, 20 to 30%, and 20 to 30% in these steps, respectively. Scott et al. [72] observed that during the fermentation of wort to which OTA and FMs (B_1_ and B_2_) were added, there was a reduction of approximately 2–13% in OTA, 3–28% in FM B_1_, and 9–17% in FM B_2_. As OTA is commonly found in cereals, beers made from cereals have a potential OTA risk. The occurrence of OTA in beer and wine at low and variable levels has been the subject of reports by some authors (from less than 0.1 μg L^−1^ to more than 1 μg L^−1^). There is a lack of consistency in the data on the proportion of polluted wine samples, and the levels of OTA in these beverages are also applied to beer. In an assessment of beers consumed in Spain, OTA was found in 100% of imported beers and 97% of national beers, with the total OTA ranging in positive samples from 5–121 µg mL^−1^ [73]. Deetae et al. [74] analysed 51 samples of 17 Asian beer brands commercially available in France for the presence of OTA, which ranged from below the LOD to 175 µg L^−1^, not exceeding the recommended level for safe consumption. In another study, OTA was found at levels ranging from 8 to 498 µg mL^−1^ in 69 beer samples (34 imported and 35 national) purchased and analysed in Spanish retail outlets (overall average = 70 µg mL^−1^; national beer average = 85 µg mL^−1^; imported beer average = 55 µg mL^−1^) [75]. Nip et al. [76] informed us that OTA in barley was transferred to beer with a reduction of 14–18%, and Chu et al. [70] found that OTA in barley was reduced by 14–28% when used to brew beer, and 70% of the OTA was degraded in the brewing process [77]. Gjerten et al. [78] found that OTA in malt was transferred to beer with a reduction of 10%. During beer brewing, significant losses of OTA (40–89%) have been observed in the grist during mashing, most likely due to proteolytic degradation, and a further 16% can be removed with the spent grains [66]. In summary, OTA losses during fermentation vary from 2 to 69% [52]. 

In a study supervised by Schothorst and Jekel [79] in the Netherlands, where 51 beers were monitored, 3 beers were found to have low levels of DON (varying from 26 µg L^−1^ to 41 µg L^−1^). The Dutch white beer had the highest level of DON (41 µg L^−1^), followed by the Belgian white and ale beers (36 µg L^−1^ and 26 µg L^−1^). None of the beers examined contained trichothecenes above the LOD (25 µg L^−1^), and none exceeded the temporary tolerance limit for DON (500 µg L^−1^). Piacentini et al. [80] analysed two MTs (DON and FM B_1_) from fifty-three different Brazilian ales (pH = 4.6, acidity = 0.3, and real extract = 5.5) and lager (pH = 4.7, acidity = 0.2, and real extract = 5.2) craft beers and found no effect on these physicochemical properties. Among the positive samples (32%), a mean of 221 µg L^−1^ was registered for DON and 105 µg L^−1^ (15% of the positive samples) for FM B_1_. This can be explained by the fact that the toxins in the barley can be influenced by environmental conditions such as the weather, the growing conditions, and the agricultural practices. In another study conducted by the same authors in 2017 on 114 Brazilian lagers, about 50% of the samples were positive for F B_1_, ranging from 202 to 1569 µg L^−1^. As for DON, none of the samples were found to contain this mycotoxin. This could be explained by the fact that the transfer of MTs to beer may depend on the infection of the crop, the technological requirements of the brewing process, and the agronomic practices [81]. A recent study showed that 26% of samples from 61 Mexican market beers (ale, lager, non-alcoholic, 4–5% vol, >5.5% vol, golden, dark, amber, craft, and industrial) were positive for MTs [82]. Of the positive samples, 87% were polluted with DON and its metabolites (3G, 3A, and 15A). This reaction may be due to the high solubility of DON in water, which can be transmitted from malt to beer. Three beers were found to be contaminated with FM B_1_, which means that maize used as an unmalted ingredient could be a source of contamination. Compared to industrial beers (16%), craft beers have a higher pollution level (56%). This result is supported by Peters et al. [83], who analysed 1000 beer samples (60% craft beer) from 47 countries for the presence of various MTs: AF B_1_, OTA, FMs, DON, ZEN, T-2, and HT-2. This study found more MTs in craft than industrial beers. As for the type of fermentation, the same 2019 study found that ale beers had higher contamination levels (42%) compared to lager beers (29%), which could be due to the adsorption of toxins on yeast cells during fermentation. In addition, 70 artisanal African sorghum beers were monitored in a 2011 study [84]. DON was present in all beer samples, with an incidence of 79% for FM B1. On the other hand, in 50 artisanal African sorghum and maize beers, the occurrence of DON was 74% and FM B_1_ was 100%. This suggests that the presence of moulds is influenced by the natural conditions for making beer in Africa, which include high humidity for storing beer. In another study by Lulamba et al. [85], common MTs found during the brewing process included AF B_1_, FM B, OTA, ZEA, and DON, which are the main MTs in beer produced in sub-Saharan Africa. In the production of beers in Europe and America, residual levels of <20% of AF B_1_, OTA, and FM B_2_ can be achieved, together with the conversion of ZEA to a less toxic compound (β-zearalenol). In contrast, >50% of DON and higher levels of FM B_1_ can be recovered in the finished beer. Adsorption is the main method used to remove MTs in brewing. In contrast, there is no significant efficient removal of MTs in traditional African beer processes. This is because mycotoxigenic fungi thrive in the prevailing environmental conditions during beer brewing. Another study was carried out on 83 samples of Italian (craft and industrial) beers for OTA and DON showed low concentrations of these MTs for both, and the levels found should not affect the health of customers [86]. A study on the occurrence of 9 MTs in 100 beer samples produced in Latvia showed that the most common MTs were HT-2 and DON, which were identified in 52% and 51% of the samples analysed, respectively. The highest level was observed for DON, which reached 248 µg kg^−1^ [87]. Table 2 shows the MTs detected in the monitoring of 220 beer samples during a five-year period (2014–2018) carried out in the Czech Republic by Olšovská et al. [88], where only 8 samples were positive for the detection of MTs and all of them were below the set limit.

The adsorption of spent grains during brewing has been shown to convert many toxins into less toxic compounds or to reduce their concentration. ZEA and PAT are two of the MTs that are metabolised during the fermentation step and therefore pose little risk of pollution in the beer. AFs (B_1_ and B_2_), together with OTA, were reduced to residual levels (<20%) during the mashing process after artificial inoculation of the raw materials. This showed that the toxins posed little health risk in the beer, as they disappeared during the rest of the brewing process [89]. In addition, a similar reduction in AF B_1_ and FM B_1_ concentrations was stated in another study [90]. This shows that MTs are not a problem in beer products if the cereal raw materials comply with the limits set by national or international regulations.

### 3.4. Nitrosamines

There are no known industrial uses for N-nitrosamines (NAs). However, as unintentional by-products of food processing, they can be found in processed foods. NAs form when nitrates or nitrites react with certain amines. Several consumer products contain NAs and/or their precursors, which include alcoholic beverages, processed meats, cosmetics, and cigarette smoke. More than 24 N-nitrosamine compounds contribute to the total N-nitrosamines (TNAs). N-nitrosodimethylamine (NDMA) and N-nitrosodiethylamine (NDEA) are the predominant NAs formed (Figure 4). When dichloramine (NHCl_2_) in water reacts with the available organic matter of natural or artificial origin, NAs can also be formed as disinfection by-products. NAs are believed to be potent carcinogens as they can cause cancer in various organs and tissues, including the brain, lungs, liver, bladder, kidney, stomach, oesophagus, and nasal sinuses [91]. 

NAs were first identified in beer and malt in the 1970s. Their proven presence in beer, together with their significant health risks, has led to the development of new technological processes, particularly in malt kilns. Following these technological changes, the concentrations of NAs in malts and beers were rapidly reduced, and, consequently, the interest of researchers in these compounds has greatly diminished in the last 15 years, although some questions concerning these compounds have still not been answered [92,93,94]. Chromatographic separation of the components of interest (mostly LC) with MS detection is used in most existing methods for the determination of NAS [95]. 

Malt is now known to be the source of NDMA in beer, although it is not clear how much NDMA was formed or removed during brewing [96]. Barley has little NDMA appearing in malt during kilning [97]. A methodical study of other beer ingredients showed that NDMA is not usually present in significant amounts in any processing aids, additives, or ingredients except malt, although small amounts were found in hops. However, the presence of NDMA in beer is negligible due to dilution (about 1 in 500 during brewing). The NDMA content in water is moderately variable, although it can reach 1 µg L^−1^, and it is not readily eliminated by boiling (no more than 20%). The most credible evidence that NDMA is not normally produced in the mash, wort boil, or fermenting process is the fact that most beers contain less than 1 µg L^−1^. Malt is known to contain a few µg L^−1^ of dimethylamine (DMA). However, it is not normally converted into NDMA when stored. It was thought that a reaction between DMA or other nitrogenous organic compounds in the malt and nitrate/nitrite in the brewing liquor could form NDMA during mashing or wort boiling. NAs have been found in wine and other non-malt beverages, and microorganisms or even inert particles are known to catalyse the generation of NDMA from DMA. Consequently, the formation of NDMA during fermentation might have been probable. Although wort and beer contain DMA, there is no evidence that beer fermentation produces NDMA. This is probably because the concentration of NO_3_^−^ is too low even for the catalysed reaction [93]. 

Out of a total of 220 samples of beer analysed in the Czech Republic between 2014 and 2018, only 1 sample of mixed beer was found to exceed the limit for NDMA, NDEA, N-nitrosopiperidine (NPRN), and TNAs in 2015, and only 1 sample of dark lager was found to exceed the limit for HIS in 2017. In the other studied years, all samples tested were in compliance with the required standards and regulations [88].

### 3.5. Pesticides

A pesticide (PC) is “any substance, or mixture of substances of chemical or biological ingredients intended for repelling, destroying, or controlling any pest, or regulating plant growth” [98]. The term includes compounds like antimicrobial, defoliant, disinfectant, fungicidal, herbicidal, insecticidal, insect growth regulator, molluscicidal, and other minority classes. Crop protection products contain both active and inert ingredients. Inert ingredients (stabilisers, colourings, etc.) are important for product usability and performance, while active ingredients are used for pest, disease, and weed control. The term “plant protection product” (PPP) is often used instead of “pesticide”, although the latter has a wider range of uses, as it can be used for purposes other than plant protection, such as the control of livestock, household, and industrial pests.

Owing to their high selectivity, good resolution, and separation efficiency, chromatographic techniques, mainly GC and LC coupled to MS, are commonly used for the analysis of PC residues in different matrices. Different ionisation sources such as electron impact (EI), chemical ionisation (CI), atmospheric pressure chemical ionisation (APCI), atmospheric pressure photoionisation (APPI), or electrospray ionisation (ESI) coupled to different analysers such as ion trap (IT), quadrupole (Q), triple quadrupole (TQ/QqQ), time of flight (TOF), and/or quadrupole time of flight (Q-TOF) are generally used. For water-soluble PCs, many analytical methods exist using chromatographic techniques other than GC, such as HPLC, supercritical fluid chromatography (SFC), ultra-high-performance liquid chromatography (UHPLC), or ultra-high-performance convergence chromatography (UPC2). A large number of analytical methods using these techniques have been recently proposed for the analysis of PC residues in wheat and barley, malt, hops, and beer [99,100,101,102,103,104,105,106,107,108].

Many PCs may be used during the growing season to ensure high quality and food safety when growing crops. The use of PCs on barley and hops allows good yields to be achieved and reduces losses during storage [109]. The most commonly used herbicides, insecticides, and fungicides in barley and hops include sulfonylureas, pyrethroids, and triazoles, respectively. The problem is that residues of these PCs in the barley can pass into the beer. However, residues can also come from the soil itself and from the water used, as water is its main component (about 90%) [7,110].

A probable source of unwanted contamination during beer production is the PCs left on the barley grain. The health hazard of barley grains containing PC residues is of particular concern to the brewing industry in many countries, in addition to the optimal physiological function of the barley to be malted. The quality of the raw materials has a critical influence on the quality of the beer and determines how they are handled and processed [111].

The extensive use of PCs on barley and hops could lead to their residues appearing in beer. The main factors (temperature, pH, and water content) influencing residue stability and physicochemical properties of PCs (octanol/water partition coefficient, vapour pressure, and water solubility) are critical to their ultimate fate. Many PCs are adsorbed on the spent grains after mashing, resulting in a decrease in unhopped wort. In addition, the concentration of these substances decreases during cooking and fermentation. In general, maltsters should pay special consideration to the residues of hydrophobic PCs since they can be left on the malt. Conversely, brewers should be on the lookout for residues of hydrophilic PCs, as they can affect the quality and beer organoleptic characteristics (flavour, taste, aroma, or colour).

As some authors have pointed out, PC residues can remain on the malt during the first step (malting) [112,113,114] (Table 3). Subsequently, during the mashing and boiling steps, PCs on the malt can be transferred to the wort in varying proportions, depending on the process used, but it should be emphasised that the removal of trub and spent grains tends to reduce the residual PCs, as most of them have a low solubility in water [106,115,116,117,118,119] (Table 4). Hakme et al. [120] studied the fate of 15 pesticide residues (13 fungicides, 1 herbicide, and 1 growth regulator) in field-treated rye, wheat, and barley samples used as an adjunct during the brewing process and concluded that, on average, 58% of the PC residues were recovered in the by-products, with 53% recovered in the spent grain, 4% in the trub, and 1% in the spent hops. No residues were found in the spent yeast, while 9% of the residues were recovered in the beer. This is consistent with the tendency of non-polar pesticides (fungicides) to remain adsorbed on spent grains during the brewing process. The most polar PCs included in this study (Mpiquat and glyphosate) showed different behaviour, with the highest proportion (>80%) being recovered in the sweet wort and moved to the beer. An excellent analysis can be found in the paper by Inoue et al. [103], which investigated the fate of 368 PC residues during beer brewing. Only a few of the PCs remained at high levels in the beer. In particular, methamidophos, which has a high solubility in water (200 g L^−1^), persists at about 80%. 2-(1-naphthyl)acetamide and imazaquin remained at 70–80%. Flumetsulam, fluoroxypyr, thiamethoxam, imibenconazole-desbenzyl, tebuthiuron, and imidacloprid remained at 60–70%. These nine PCs were largely retained in the unhopped wort (log *K*_OW_ < 2) according to their physical properties. Log KOW is frequently used in environmental studies as an indicator that a PC will be subject to bioaccumulation. Finally, when PC residues, especially some fungicides, are dissolved in the brewing wort, some organoleptic changes may occur in the finished beer, which may have hazardous effects for the consumer [99,121,122,123,124,125,126]. For more detailed evidence on the behaviours and fate of PT residues during brewing, see the recent paper by Pérez-Lucas et al. [110].

### 3.6. Acrylamide

Acrylamide (AA) is a chemical that may form in some foods from carbohydrates and asparagine (an amino acid) during heat cooking, including frying, roasting, and baking. [127]. It does not come from the packaging of the food or the environment. It is likely that AA has always been present in cooked food. However, acrylamide was first detected in certain foods in April 2002 [128,129]. AA (or acrylic amide) is a vinyl-substituted primary amide with the chemical formula CH_2_ = CHC(O)NH_2_ (Figure 5). This compound is one of the substituted olefin monomers, commercially available since the mid-1950s, tested for carcinogenicity in the early 1970s of unusual cancers, particularly liver angiosarcomas, in factory workers exposed to another important monomer (vinyl chloride). In the 1980s, several studies were published on the carcinogenicity of AA in mice and rats. AA caused tumours at multiple sites in both mice and rats when administered systemically by various routes in these experiments, which included a 2-year bioassay in rats and several shorter-term studies in mice. AA is converted in vivo to its epoxide, glycidamide, which is genotoxic in various in vitro and in vivo test systems [130].

**Table 4 foods-13-01709-t004:** Residues of various pesticides remaining after crushing and boiling (%).

Pesticide	Log *K*_OW_	Sweet Wort	Spent Grains	Brewer Wort	Spent Hops	References
Atrazine	2.5	45	55	42	20	[115]
α-BHC	4.0	8	54	30	15	[116]
Captafol	3.8	BDL ^a^	3	BDL	BDL	[116]
Chlorpyrifos	4.7	17	3	4	32	[116]
Cyproconazole	3.1	10	40	9	ND ^b^	[117]
Deltamethrin	4.6	BDL	45	3	37	[116]
Dichlorvos	1.9	8	BDL	BDL	BDL	[116]
Diclofuanid	3.7	10	10	BDL	BDL	[116]
Dicofol	4.3	BDL	70	18	60	[116]
Diniconazole	4.3	4	49	3	ND	[117]
Epoxiconazole	3.4	8	44	7	ND	[117]
Fenitrothion	3.4	4	30	3	ND	[118]
Fenobucarb	2.8	35	30	64	1	[116]
Fenvalerate	5.0	BDL	50	3	7	[116]
Flucythrinate	6.2	BDL	60	BDL	10	[116]
Flutriafol	2.3	13	36	10	ND	[114]
Glyphosate	−3.2	97	3	95	2	[116]
Malathion	2.7	20 7	35 40	15 4	5 ND	[116,118]
Myclobutanil	2.9	9	38	8	ND	[117]
Nuarimol	3.2	6	26	6	ND	[117]
Oxamyl	0.4	1	BDL	20	BDL	[116]
Parathion-methyl	3.0	1	BDL	10	3	[116]
Pemdimethalin	5.2	1	21	1	ND	[118]
Permethrin	6.1	BDL	70	BDL	50	[116]
Pirimicarb	1.7	84	14	50	3	[116]
Pirimiphos-methyl	4.2	2	68	6	12	[116]
Propiconazole	3.6	4	42	4	ND	[117]
Tebuconazole	3.7	8	44	7	ND	[114]
Terbutylazine	3.2	12	80	7	40	[115]
Triadimenol	3.1	36	ND	ND	ND	[131]
Trifluralin	5.3	1	17	1	ND	[118]

^a^ Below detection limit; ^b^ not determined.

In its 2015 risk assessment, EFSA concluded that the current levels of dietary exposure to acrylamide do not pose a health concern, although they may be a concern for young children with high dietary exposure [132]. Commission Regulation (EU) 2017/2158 [133] establishes mitigation measures and benchmarks to reduce the presence of AA in foods. The EU benchmark levels vary depending on the type of food and can range from 40 µg kg^−1^ in baby food to 4000 µg kg^−1^ for coffee substitutes derived exclusively from chicory. To support actions to reduce AA levels, large food manufacturers are expected to conduct representative sampling and analysis to evaluate mitigation measures. The US Food and Drug Administration (FDA) has issued guidance outlining current recommendations for reducing AA levels in certain foods. The FDA guidance recommends that food companies be aware of the levels of AA in their products and, where feasible, implement approaches to reduce these levels. Both GC-MS and LC-MS are recognised as the main, useful, and authoritative methods for determining AA in food and drink [134]. Owing to the complexity of processed food samples, sample extraction and clean-up can often involve labour-intensive methods. Prior to GC analysis, samples also require derivatisation, which is often achieved by bromination. LC-MS methods do not require this step, saving analysts time. 

Malt is usually kilned at temperatures for the given malt type (up to 225 °C) for 90 min to 2.5 h. Mikulíková et al. [135] showed that the maximum AA formation was detected in the thermal interval 150–170 °C. Thereafter, AA formation decreases. The decrease in AA formation at higher temperatures can be explained because AA, as an intermediate product of the Maillard reaction, continues to react, and this non-enzymatic browning reaction produces other compounds [136]. In pale malts, the AA content ranged from 630 to 660 µg kg^−1^. The AA content in special melanoidin malt was 2210 µg kg^−1^. Melanoidin malt has the same AA content as malt sampled during kilning at 130 °C. This high AA content corresponds to the production conditions of melanoidin malt, which favour the formation of Maillard reaction compounds during kilning. Compared to barley malts produced at the same temperatures, rye malts have lower AA levels in caramel and roasted malts. The lower AA is caused by different levels of asparagines and reducing sugars in modified rye malt. The lower content of AA in wheat malt can be explained in a similar way. Among the roasted barley malts, the highest AA content was found in the Carafa^®^ special malt. The highest AA content of all malts analysed was found in Caramel malt (3.1 mg kg^−1^). This high level of AA is consistent with the method used to produce caramel malt (kilning temperature 150–170 °C) and with the observed thermal dependence of AA formation during kilning. Despite the high level of AA in malt, its content was below the LOD (<25 µg L^−1^) in all beer samples analysed. This fact seems to be related to the beer production process, where 2–4 hL of water per 100 kg of grist are used for mashing. Similarly, no detectable levels of AA were found in beer in a study carried out in Brazil to assess AA levels in various foods [137].

The formation of AA is slightly more dependent on temperature than the formation of colour. This means that a reduction in temperature will have a greater effect on AA formation than on colour formation, with a negative effect on the perception of taste, texture, and/or colour [138]. In the study by Bodagnova et al. [87], AA was detected in beers at a mean concentration of 2.3 μg kg^−1^, with a range of 0.5–14 μg kg^−1^. Few publications are available on the occurrence of AA in alcoholic beverages sold in the EU. Concentrations were below the LOD (5 μg kg^−1^) in the majority of beer samples analysed. For example, none of the samples analysed contained AA above the LOD in an analysis of beer sold on the Swedish market [139]. Furthermore, beer was reported to be significantly less polluted with AA than any other food analysed [140]. The author also found that beer colour correlated with the presence of AA, whereby it was more common in pale or medium-coloured beers, probably due to reduced AA evaporation at higher roasting temperatures. This tendency is also reflected in the study by Bodagnova et al. [87], where the occurrence of AA in pale beers was 80%, while only 54% of dark beers had AA pollution. The mean concentration of AA in these beers also followed the same trend, being lower in the dark beers (1.5 μg kg^−1^) than in light beers (2.2 μg kg^−1^). The highest AA level ever found in beer was found in German wheat beer (72 μg kg^−1^). However, of the 11 German beers analysed in this study, this was the only sample to contain AA at a detectable level [139]. There are currently no EU or international regulations specifying the acceptable levels of AA in beers and other alcoholic beverages. Meanwhile, the guide values for other products based on rye, wheat, maize, spelt, oats, barley, and rice (300 and 200 μg kg^−1^) are much higher than the concentrations set out in EU Commission Recommendation 2013/647 [141], in comparison with the recommendations for cereal-based foods from the EFSA summary of surveillance data [132].

### 3.7. Micro- and Nanoplastics

Microplastics (MPs) are defined as “solid particles containing polymers, to which additives or other substances may have been added, of all dimensions 100 nm ≤ x ≤ 5 mm” [142,143]. The definition of microplastics therefore includes particles from the millimetre scale down to the smaller nanoscale. Nanoplastics (NPs), on the other hand, are particles ranging from 1 nm ≤ x ≤ 1 μm [144]. They may be either intentionally produced at this size (primary microplastics) or result from the fragmentation of any type of plastic (secondary microplastics). Plastics are part of our daily lives, and globally, we use 4 trillion plastic bags per year and 1 million plastic bottles per minute. MPs are considered emerging pollutants of high environmental concern that are increasingly detected and quantified, especially in aquatic environments. Most plastic waste takes up to 500 years to decompose and is not biodegradable [145]. While the effects of MPs on the environment are under investigation, there has yet to be comprehensive monitoring of MPs in food and the impacts on human health. The type of plastic and its chemical composition, as well as absorbing and releasing plastic-affinity chemical contaminants, also affect the effects of MPs. They contain additives such as phthalates and bisphenols that can be absorbed by organisms and enter the food web. Fibrous MPs in the air can get into our respiratory system and pose a risk to the environment and human health [146]. 

MPs are composed of a variety of polymer types (Figure 6). The most produced and consumed are polypropylene (PP), low- (LDPE) and high-density polyethylene (HDPE), polyvinyl chloride (PVC), polyurethane (PU), polyethylene terephthalate (PET), polystyrene (PS), and others such as polycarbonate (PC), polyamide (nylon) (PA), polymethyl methacrylate (PMMC), and polytetrafluoroethylene (PTFE) (Figure 6). Global plastics consumption follows the order PE > PP > PVC > PET > PS [147]. They come in a wide variety of sizes, colours, shapes, and material types and come from a wide range of sources.

Quantifying environmental MPs is not straightforward. It is therefore important to note that there are special considerations to be taken into account when determining residual contaminants, and that the sampling strategy will be influenced by the objective of the study. To date, there is no standardised method for any of these purposes. However, progress is being made in this direction [148]. The characterisation of MPs is an analytical challenge due to the lack of harmonisation of the offered analytical methods. This makes it difficult to compare different studies [149]. The simplest method of MP characterisation for most researchers is a visual inspection. With the naked eye, coloured plastic fragments in the range of 2–5 mm can be identified. Analysis is more difficult when the particles are ≤1 mm [150]. 

In order to identify and/or quantify MPs, different analytical techniques based on spectroscopy, microscopy, and/or thermal analysis have been used. The use of spectroscopic techniques such as Fourier Transform Infrared (FTIR) and Raman spectroscopy is the most common characterisation approach described in the literature. In addition, scanning electron microscopy (SEM), scanning electron microscopy-energy dispersive X-ray spectroscopy (SEM-EDS), and, less commonly, environmental scanning electron microscopy-energy dispersive X-ray spectroscopy (ESEM-EDS) have also been used to characterise MPs [151]. However, these techniques can be very time-intensive. In addition, because the isolation of MPs depends on the skill of the researcher, chemical characterisation can be subject to selection bias [152]. In contrast to these techniques, thermal analysis of MPs is a completely different approach that is increasingly being used for their characterisation [153]. The method is based on identifying the polymer according to the degradation products it produces. Thermal analysis includes various techniques such as pyrolytic gas chromatography-mass spectrometry (py-GC-MS), thermal gravimetry (TGA), hyphenated TGA such as TGA-mass spectrometry (TGA-MS), TGA-thermal desorption gas chromatography-mass spectrometry (TGA-TD-GC-MS), and TGA-differential scanning calorimetry (TGA-DSC). For the characterisation of low-solubility MPs and additives that are not readily soluble, extractable, or hydrolysable, the development of thermal methods is fundamental. Furthermore, chromatographic techniques play a very important role, especially LC, combined with different detection systems.

Liebezeit and Liebezeit [154] analysed 24 samples of German beers. Germany beers presented the highest MPs number, compared with Mexican and American beers. They identified the presence of fragments, fibres, and granules of PA, PEA, and PET (size range = 1–2 mm). The values found ranged from 2 to 79 fibres L^−1^, 12 to 109 fragments L^−1^, and 2 to 66 granules L^−1^. By means of identification by Raman spectroscopy, the presence of PS and PP in beers was observed by Li et al. [155]. MPs, which could be divided into fibres and fragments, were observed in all beers from different countries. They were visually identified by their colour and structural characteristics. Several beers contained microplastic fibres longer than 100 µm. The number of microplastic fragments in each sample was estimated to be between 929 and 9154 per 100 mL of beer, depending on the volume. As a minor component, fibrous MPs account for an average of 24.6%. In total, Tempt 7 (Denmark) has 1212 microplastics per 100 mL, while Barbarella (Brazil) has as many as 9659 per 100 mL. The number of MPs per 100 mL of beer ranged from 1212 to 9659, indicating multiple microplastic exposures.

#### Chemical Pollutants Related to Plastics (Phthalates and Bisphenols)

Phthalates (PTs) and bisphenols (BPs) are organic compounds used in combination with other chemicals in the manufacture of certain plastics and resins. Based on scientific evidence of endocrine disruption, persistence, production volume, and potential exposure risks, several reports have identified a list of priority chemicals [156]. In particular, PTs and BPs used as plasticisers have been included in category 1, based on a substance’s endocrine disrupting activity in at least one species.

PTs are used to manufacture polyvinyl chloride (PVC) plastics. Food is the main vehicle for human exposure to PTs, which are often present in packaging materials or during food processing [157]. PTs are diesters of phthalic anhydride obtained by its reaction with oxoalcohols to form esters. This occurs by (i) alcoholising the phthalic anhydride to develop the monoester (an irreversible and fast reaction) and (ii) converting the monoester to a diester (a reversible reaction), which usually needs a catalyst. Depending on the nature and length of the oxo alcohols (C1–C13) from which they are made, there is a wide range of PTs available [158]. They can be divided into high-molecular-weight (HMW) and low-molecular-weight (LMW) PHs (Figure 7). HMW PTs are characterised by lower bioaccumulation factors, including di(2-ethylhexyl) phthalate (DEHP), the most commonly used phthalate plasticiser for PVC [159], butylbenzyl phthalate (BBzP), diisononyl phthalate (DiNP), di-n-octyl phthalate (DnOP), and diisodecyl phthalate (DiDP). On the other hand, LMW PHs include di-butyl phthalate (DBP), dimethyl phthalate (DMP), diethyl phthalate (DEP), and di-isobutyl phthalate (DiBP). Contrarily, the LMW are characterised by higher bioaccumulation factors, with DBP being the most studied [160].

The hydrolytic cleavage of diesters is the first step in the metabolic transformation of PTs after exposure [161]. Hydrolysed monoesters from this molecular step are then eliminated in urine or further processed as glucuronide conjugates, some of which are oxidised. The main metabolites of PTs are di-2-ethylhexyl phthalate (DEHP), monoethyl phthalate (MEP), monobutyl phthalate (MBP), mono-benzyl phthalate (MBzP), mono(2-ethylhexyl) phthalate (MEHP), mono(2-ethyl-5-hydroxyhexyl) phthalate (MEHHP), mono-(2-ethyl-5-oxohexyl) phthalate (MEOHP), mono-(2-ethyl-5-carboxypentyl) phthalate (MECPP), mono-isobutyl phthalate (MiBP), and mono-(3-carboxypropyl) phthalate (MCPP) [162]. Many of them are endocrine disruptors compounds (EDCs), “*chemicals or mixtures of chemicals that interfere with some aspect of hormone action*” [163]. In general, EDCs can disrupt the endocrine system by competing with endogenous steroids for binding to receptors and hormone transport proteins. So, the metabolism or synthesis of endogenous hormones may also be altered. This ultimately affects the recruitment of transcription factors and alters how the cells express genes [160,164]. The EU has published a list of chemicals with a proven or potential effect on the endocrine system, such as DBP, BBzP, and (DEPH) according to Regulation (EC) No. 1907/2006 [165]. 

To date, several methods for the determination of PTs in different alcoholic beverages have been developed in the literature. In particular, Leibowitz et al. [166] developed a method for the determination of PTs in grain neutral spirits and vodka without sample preparation or sample enrichment using LC and GC coupled with MS, although this method cannot be used for sugar-containing alcoholic beverages. In addition, Del Carlo et al. [167]. developed a method for the determination of PTs in wine using solid-phase extraction and gas GC-MS, and Gao et al. [168] analysed trace PTs in beer by solid-phase microextraction and GC. Russo et al. [169] developed a method for light alcoholic beverages and soft drinks using XAD-2 adsorbent and GC coupled with ion trap mass spectrometry (ITMS) detection. More recently, Gemenetzis et al. [170] proposed HPLC-UV detection for the analysis of DEHP.

PTs can be a source of contamination for barley and hops through dust and rain deposition. They can also contaminate as soon as they are harvested and placed in plastic bags to transport to manufacturing plants. However, alcoholic beverages, especially those with a high ethyl alcohol content, may not be packaged in plastic containers, although alcoholic beverages sometimes contain high levels of PTs, especially DEHP. This may be due to the plastics and raw materials used to manufacture the product [171]. For beer and wine, phthalate contamination can come from plastic gaskets, lids, and stoppers [172], but also from tetrapacks, cans [173], and bottles, depending on how they are stored. During the brewing and bottling of beer, the liquids come into contact with plastic tubes and connections. They are also stored in considerable containers for a period of time during production, which may also be aged in steel or coated concrete tanks. These are likely to be supplementary sources of pollution [174].

Olšovská et al. [88] investigated the presence of two PTs (DEHP and DBP) in 220 beer samples over a five-year period (2014–2018) and concluded that these PTs were below the LOD (10 µg L^−1^) in all cases. Ye et al. [175] detected the presence of DBP and DEHP in bottled beer samples, with total PT concentrations ranging from 6.2 to 7.8 µg L^−1^. The migration test showed that the high content of DEHP included in PVC seals in lids could be a potential source of PT pollution in bottled beer during transport and storage. In another study, the levels of various PTs (DMP, DEP, DiBP, DBP, DEHP, and DnOP) were monitored in beers packaged in aluminium cans, PET, and glass bottles. A total of 10 beers packaged in aluminium cans, 16 beers packaged in PET, and 18 beers packaged in glass bottles were purchased from a local grocery store. The results indicate that the presence of PTs in beers in PET packaging may be significant. In one sample, the total sum of PTs reached 220 µg L^−1^. In particular, high concentrations of DBP were found in all samples, with the highest concentration reaching 92 µg L^−1^. However, canned beer contained even higher levels of certain PTs, such as DEHP, which reached 327 µg L^−1^ in one sample [176]. Overall, the results indicate that beer in cans has higher levels of total PTs than beer in PET bottles. Since many manufacturers use cans, this must be taken into account. However, canned beers had high levels only of MEHP, whereas PET-bottled beers had high levels of DiBP, DBP, and MEHP. This could be due to a weaker bond between the PET matrix and the PTs, resulting in their easier release in PET-bottled beer. Beer in glass bottles, which contain the lowest levels of PTs, appears to be the safest from a health perspective. In a study conducted by Pereira et al. [177] to simultaneously assess the levels of six PTs (BBP, DBP, DEP, DEHP, DIBP, and DMP) in 66 commercial beers, five of the six compounds studied were found, with levels ranging from 2 to 205 μg L^−1^. The most abundant was DEHA (205 μg L^−1^), while DMP was not present in any sample. The highest levels of these pollutants were found in samples containing 5–6% alcohol, packaged in aluminium cans, and produced in an industrial environment. Despite the low risk of exposure to PTs from beer, the ubiquitous nature of these compounds must not be forgotten. This may lead to cumulative exposure.

On the other hand, bisphenols (BPs) are a group of compounds that have a similar chemical structure, consisting of two phenolic rings linked by a carbon that is connected to several side groups. They include bisphenol A (BPA), bisphenol B (BPB), bisphenol AP (BPAP), bisphenol F (BPF), bisphenol AF (BPAF), bisphenol S (BPS), bisphenol Z (BFZ), and bisphenol P (BPP). BPA ([4,4′-(Propane-2,2-diyl]diphenol) (Figure 7) and its derivatives are the main constituents of PC plastics and epoxy resins and have been widely used in the manufacture of packaging materials such as storage containers, baby bottles, oven bags, and metal cans [178]. As mentioned above, BPA and other BPs are used as monomers in the production of plastic materials and are likely to be detected in food contact materials (FCMs). The largest single use of BPA is as a co-monomer in the production of PC plastics (accounting for 65–70% of the total production of BPA). The production of epoxy and vinyl ester resins accounts for 25–30% of BPA use. The remaining 5% is used as the main ingredient in several high-performance plastics and as a minor additive in polyvinyl chloride(PVC), polyurethane (PU), and several other materials [179]. It is not a plasticiser, although it is frequently mislabelled as such [180]. There has been a longstanding public and scientific debate about the effects of BPA on human health [181]. BPA has hormone-like properties that mimic the effects of oestrogen in the body. However, taking into account the toxicity studies of BPA and its analogues, their migration from plastic FCMs into food and beverages may have an adverse effect on the organoleptic properties of the products and may have a detrimental effect on human health, in particular if the consumer is exposed to higher levels of BPs than those set by the legislation [173]. BPA-free plastics made with alternative BPs such as BPS and BPF have also been introduced, but there is also debate about whether these are actually safer [182]. This is the reason why the determination of BPA and other BPs in food and beverage samples because of their migration from FCMs is of great importance. Among the different methods used to determine BPs, liquid–liquid (LLE) or solid-phase extraction (SPE) followed by GC-MS, LC-MS, or LC-FD are normally used for their analytical determination in canned food [183]. 

A Health Canada Department study measured BPA levels in different samples of canned soft drink and beer products [184]. BPA was detected in 20 out of 38 soft drink and beer products. In 18 products, BPA levels were below the LOD (5 ng L^−1^). BPA was not noticed in any of the glass soft drink samples and was only detected in one of the PET soft drink samples at a level of 18 ng L^−1^. However, low levels of BPA (19 to 210 ng L^−1^) were identified in all canned soft drink samples analysed. Low levels (81 to 540 ng L^−1^) of BPA were also found in all canned beer samples examined and only in one bottled beer sample with a level of 54 ng L^−1^. The presence of BPA in can samples and the absence (or lower concentrations) of BPA in bottle samples suggest that migration from can coatings may be the source of BPA in can beers.

### 3.8. Other Minority Pollutants

In this group, polychlorinated biphenyls (PCBs), chlorinated aliphatic hydrocarbons (CAHs), polycyclic aromatic hydrocarbons (PAHs), carbonyls, furans, and trihalomethanes (THMs) are included (Figure 8). The scientific literature on the evolution of these pollutants during beer brewing is scarce. Only a few papers have been published on their presence in raw materials and their evolution during brewing.

#### 3.8.1. Polychlorinated Biphenyls

Polychlorinated biphenyls (PCBs) are highly carcinogenic compounds. They were mainly used as electrical insulation fluids in capacitors and transformers, and as hydraulic, heat transfer, and lubricating fluids [185]. PCBs (C12H10-xClx) were used as plasticisers and flame retardants, mixed with other chemicals, in a range of products including adhesives, caulks, plastics, and carbonless copy paper. In developed countries, PCBs were manufactured from 1930 until the mid-1970s, although many continued to be used for decades after PCB production ceased. Of the 209 different types of PCBs, 13 have dioxin-like toxicity. The degree of chlorination determines their persistence in the environment with half-lives ranging from 10 days to 1.5 years. Research has linked PCBs to reproductive failure and suppression of the immune system in several wildlife species, including seals and mink. The results of the 2001 Stockholm Convention on Persistent Organic Pollutants (POPs), i.e., organic compounds characterised by long-range transport, persistence, bioaccumulation and high toxicity, resulted in a list of 12 chemicals to be regulated, nicknamed the “*dirty dozen*” [186]. Among these chemicals, PCBs have been identified as EDCs [187]. They are mainly analysed by means of GC-MS [188]. LC-MS, where the resolving power of LC and the detection specificity of MS provide high sensitivity in the analysis of PCBs, is another popular technique for the analysis of PCBs. In addition, unlike GC-MS, LC-MS does not require time-consuming derivatisation for the analysis of non-volatile compounds [189].

Thabit et al. [190] monitored the contamination of different samples of wheat and barley grains of European origin for various contaminants. None of the 14 dioxin-like PCBs (77, 81, 126, 169, 105, 114, 118, 123, 156, 157, 167, 189, 170, and 180) with non, mono, and di ortho structures, which are the most toxic congeners, were detected. They have a structure similar to PCDDs and PCDFs and action like PCDDs. The congeners detected were 28 (Russian and Latvian wheat), 52 (Ukrainian and French wheat and barley), 101 (German wheat and barley), 153 (Latvian, Lithuanian and Polish wheat), 178 (Romanian wheat), and 206 (Lithuanian wheat). Two congeners (70 and 95) were detected in some samples of Ukrainian and Latvian barley, respectively. Other congeners (138, 183, and 195) were detected in some samples of French, Estonian, and Polish wheat, respectively, and congener 187 was noticed in some samples of Russian and Estonian wheat, respectively. For PCBs in cereals, no maximum residue levels (MRLs) have been set in the EU. However, a limit of 40 ng g^−1^ for the sum of the most accumulative congeners (28, 52, 101, 138, 153, and 180) is estimated for the closest comparable commodity, i.e., vegetable oils and fats. Similar results were obtained by Witczak and Abdel-Gawad [191], who showed that congener 153, together with several other congeners, was detected in the samples of wheat and rye analysed in Poland, although at very low levels.

#### 3.8.2. Aliphatic and Aromatic Hydrocarbons

Hydrocarbons are an important source of environmental pollution. Chlorinated aliphatic hydrocarbons (CAHs), such as tetrachloroethane (TeCA), perchloroethylene (PCE), and trichloroethylene (TCE), among others, are ubiquitous pollutants that persist in the environment for many decades, mainly due to the physico-chemical properties of these compounds. They cause serious damage to human health and the environment due to their high toxicity, bioaccumulation, resistance to degradation, carcinogenicity, teratogenicity, and mutagenicity [192]. Due mainly to long-term anthropogenic pollution, polycyclic aromatic hydrocarbons (PAHs) are widespread throughout the world. Due to their inherent properties, such as heterocyclic aromatic ring structures, hydrophobicity, and thermal stability, PAHs are recalcitrant and highly persistent in the environment [193]. PAH contaminants have been shown to be highly toxic, teratogenic, carcinogenic, mutagenic, and immunotoxic to a broad range of lifeforms. PAH are organic contaminants consisting of two or more condensed aromatic rings of carbon and hydrogen atoms (Figure 8). As a function of the number of rings present in the compounds, PAHs are divided into (i) low-molecular-weight PAHs (LMW PAHs) with two or three aromatic rings and (ii) high-molecular-weight PAHs (HMW PAHs) with four or more aromatic rings. Depending on their molecular weight, they are either gaseous (LMW PAHs) or particulate (HMW PAHs). Due to their structure, PAHs have low water solubility, low vapour pressure, and high melting and boiling points. As the molecular weight of PAHs increases, there is a tendency for a decrease in water solubility and an increase in lipophilicity, making them more recalcitrant compounds [194]. Sources of PAH pollution can be segregated into two main types: anthropogenic (domestic, mobile, agricultural, and industrial) and natural (natural forest fires, volcanic eruptions, and lightning-caused peatland fires) [195]. For living organisms, including micro-organisms, animals, and humans, many PAHs are mutagens, carcinogens, teratogens, and immunotoxins [196].

The main source of CAHs in beer can be either groundwater or surface water. The technological treatment of drinking water in breweries may, in some cases, produce these compounds [88,197]. Among PAHs, benzo[a]pyrene (BaP) was not noticed in any sample analysed, whereas benz[a]anthracene (BaA) was found in some wheat samples from France, Latvia, Estonia, and some barley samples from Ukraine and Russia. Benzo[b]fluoranthene (BbF) was detected in some Russian wheat and barley samples and in some Ukrainian and Polish wheat samples. Chrysene (CRS) was detected in some Romanian and German wheat samples and in some Latvian barley samples. Anthracene (ANT) and fluoranthene (FRT) were detected only in a few samples of wheat from Germany and Lithuania, respectively, while phenanthrene (PNT) was detected only in a few samples of wheat from Estonia and Russia. Anthracene (ANT) and fluoranthene (FRT) were detected only in a few wheat samples from Germany and Lithuania, respectively. Phenanthrene (PNT) was identified only in a few wheat samples from Russia and Estonia. The results showed that PAHs were found in most of the beer samples analysed. The total sum of PAHs ranged from 6.4 to 27 μg L^−1^ and the highest PAH4 content was 2.8 μg L^−1^. Only three beer samples were PAH-free.

#### 3.8.3. Carbonyl and Furan Compounds

Carbonyl (CB) pollutants with toxic potential effects such as acetaldehyde (AD), acrolein (AC), urethane (UT), and formaldehyde (FD) and furan (FR) compounds such as furan (FR), benzofuran (BFR), tetrahydrofuran (TFR), furfural (FF), and furfuryl alcohol (FA) may be present in beer [198,199]. Their presence may vary depending on the brewing process. CBs come from both primary emissions (vehicle exhaust, solvent volatilisation, industry, and plants) and secondary production (formed by the photooxidation of volatile organic compounds (VOCs) from anthropogenic and natural sources [200]. They are low-molecular-weight, highly volatile compounds found in heat-treated commercial foods and are formed by the thermal degradation of natural food precursors such as amino acids, carbohydrates, ascorbic acid, carotenoids, and unsaturated fatty acids. Due to their electrophilic nature, they are highly reactive. As a result, they readily react with biological nucleophilic targets such as proteins, RNA, and DNA. On the other hand, FR (C4H4O) is a five-membered heterocyclic compound with an oxygen-containing unsaturated ring. Compounds containing the FR ring are usually referred to as FRs, also including 2-methylfuran, 3-methylfuran, and 2,5-dimethylfuran. There is strong evidence that FR carcinogenesis involves indirect mechanisms such as epigenetic alterations, oxidative DNA damage, and regenerative hyperplasia, all of which involve tissue damage. Its half-life is short, being metabolised to the reactive metabolite cis-but-2-en-1,4-dialdehyde (BDA) by cytochrome P450 2E1 (CYP2E1), which is capable of covalently binding to amino acids, proteins, and DNA, with cell and tissue damage, mitochondrial dysfunction, and fibrosis, particularly in the liver, being the ultimate consequences of BDA binding [201].

IARC classifies AD and FD as carcinogenic (Group 1) to humans, while UT and FA are classified as probable human carcinogens (Group 2A) and possibly human carcinogens (Group 2B), respectively [202]. AC and FF are not classifiable as carcinogenic to humans (Group 3), which includes compounds for which further research is needed before they can be classified as carcinogenic. FR, BFR, and TFR are also classified as category 2B. Due to their low concentrations, high reactivity and volatility, and the presence of other major compounds, the determination of these compounds is challenging. They have been analysed in beers mainly by GC-MS or HPLC-DAD using, among others, SPE, HS, and HS-SPME as extraction methods [198]. The same authors conducted a study using 30 samples of beers ale (8 ale and 22 lager) beers from different brands, where UT was detected in only 2 samples (1 lager and 1 ale). FF was found in 37% and 82% of the ale and lager beers, respectively, while AD, AC, and FD were found in all samples. However, AC was the only CB compound found in the commercial samples in concentrations that may present a health risk. In addition to FF and FA, four other furan-containing compounds (acetylfuran, γ-nonalactone, 5-methyl-2-furanmethanethiol, 5-methylfurfural) were found in beers, although at levels too low to be a potential health hazard. Hernandes et al. [199] found AD, AC, FD, and FA residues in all brewing stages of ale and lager craft beers, while UT and FF levels were below the LOQ (0.1 µg L^−1^). The differences in raw material composition, boiling time, and fermentation temperature are responsible for the observed differences in target compound levels between the lager and ale brewing stages. Boiling and fermentation in ale brewing appear to be important steps in AC and AD formation, respectively, with boiling leading to increased FA in both beers. FF is formed by a Maillard reaction during kilning, and reducing this compound during brewing increases the FA content [203]. Boiling seems to play an important role in the formation of FA. The levels of this compound after boiling were notoriously higher than in the previous step (mashing) and in the subsequent steps (fermentation, maturation, and pasteurisation). Higher levels of AD in ale (15 μg L^−1^) than in lager (1.3 μg L^−1^) indicate that fermentation conditions (temperature and yeast strain) have an important effect on the formation of this compound. Similar results were obtained by Webersinke et al. [204], who also confirmed that fermentation temperature had a notorious effect on the AC level in ale beer, as its level was significantly higher when the wort was fermented at 26 °C (14.4 mg L^−1^) compared to 14 °C (3.7 mg L^−1^). The occurrence of AC may be indicative of environmental contamination of the grain used to produce the malt and/or the formation of this compound during the thermal steps of malting. AC may also be formed from the thermal degradation of carbohydrates, amino acids, and triglycerides during mashing and boiling when malt is heated [205]. On the contrary, pasteurisation and maturation decreased the levels of these pollutants in both types of beer. Due to the electrophilic character of aldehydes (AC, FD, and AC), they may bind to phenolic compounds, which could explain the reduction of AC levels during ale maturation [206]. No increase in AC concentration was found in the lager brew, probably due to the difference in the boiling time (60 and 90 min for ale and lager, respectively) between these two types of beer. Few studies have been devoted to the determination of FR in beer samples due to the typically low levels of pollution. According to Bodagnova et al. [87], FR was present in a total of 100 beers made in Latvia and ranged from 1.4 to 32 μg kg^−1^, with 9.5 μg kg^−1^ as the mean. The beer was found to have a relatively low level of contamination compared to other foods that may contain FR. In another study, FR was found in 102 samples of beer at levels of up to 28 μg L^−1^ in surveys carried out between 2006 and 2010, as summarised by EFSA [207]. In 13 different beers containing malt, hops, and rice, the analysis of FR showed a maximum level of 15 μg L^−1^ [208].

#### 3.8.4. Trihalomethanes

Water is a key ingredient in the brewing process. A reliable supply of high-quality water is an essential part of the beer brewing process. The properties of water that define its quality vary depending on the source of the water—municipal, surface, or groundwater [209]. All water should be free of waterborne organisms, such as bacteria, that come into contact with the product stream. Generally, most brewing water is heated and boiled in the kettle before mashing, ensuring that it is free of microorganisms. However, water may not be microbiologically sterile if it is used to dilute wort or beer or to rinse equipment. Water can be treated physically using ultraviolet light and sterile filtration, or chemically using ozone (O_3_) and chlorine dioxide (ClO_2_), a strong oxidant used as an aqueous solution. It is effective against a wide range of beer spoilage organisms, such as bacteria, yeasts, and moulds. The properties of ClO_2_ make it an ideal choice for most brewery operations. It does not form trihalomethanes (THMs), which is very important to the brewer as these compounds are extremely detrimental to beer, not to mention toxic to humans and the environment [2,210,211]. 

The three largest classes by weight of disinfection by-products (DBPs) in chlorinated water are THMs, haloacetic acids (HAAs), and haloacetonitriles (HANs) [212]. Among these, THMs are the most important class. They mainly include trichloromethane (TCM), tribromomethane (TBM), bromodichloromethane (BDCM), and dibromochloromethane (DBCM). In some cases, their amount exceeds the Maximum Allowable Concentration (MAC) in many countries around the world, causing harmful effects on human health [213]. THMs are monocarbon-substituted halogens (CHX_3_), where X can be bromine, chlorine, fluorine, or iodine or a group of these. Only TCM (chloroform) and brominated THMs such as TBM (bromoform), BDCM, and DBCM, known as total THMs (TTHMs), are significant in terms of drinking water contamination. Valdivia-García et al. [214] showed how climate can influence THM formation, pointing to the strong seasonal relationship between THM levels, dissolved organic carbon (DOC) and water temperature. The most common disinfectant is chlorine (used as an oxidant for 100 years), which is cost-effective, easy to use, and persistent in water supply networks [215]. In addition to the chlorination parameters (time and dose of chlorine), other factors such as temperature, pH, the presence of other ions (Br- and I-) or natural organic matter (NOM), chemical properties such as aromaticity and functionality, and the nature of the source water are known to affect DPB levels [213,216]. The formation of brominated THMs is justified by the presence of Br and dissolved organic matter (DOM) in drinking water [215]. Due to the fact that brominated THMs are more dangerous than their chlorinated analogues, their presence in drinking water is a major concern [217]. Although other activities such as bathing, cooking, showering, etc. are important sources of exposure due to volatilisation, the main risk to the public from THMs is direct ingestion of drinking water. The potential adverse human effects of THMs led the US EPA to establish a MAC for THMs of 80 μg L^−1^ in drinking water [213,218,219]. Likewise, the EU has set a MAC of 100 μg L^−1^ [220]. Various analytical techniques are commonly used to detect THMs in water samples. These include static/dynamic headspace, headspace/solid-phase microextraction, direct aqueous injection, liquid–liquid extraction, or membrane-based sampling, with GC coupled to MS or electron capture detection (ECD) being the most commonly used detectors [221].

The routes of human exposure to THMs include inhalation, ingestion, and dermal absorption, with volatilisation playing an important role in all three routes [222]. According to Pérez-Lucas et al. [223], the proportion of THMs eliminated after 4 h of stirring in the dark varied from 86% (TCM) to 54% (TBM), which is directly related to their Henry’s law constants and vapour pressures (TCM > BDCM > DBCM > TBM). The amount of THMs has little effect on volatilisation rates, although a linear correlation between temperature and volatilisation rate was observed. For this reason, it is very unlikely that THMS will occur in the beer, as temperatures greater than 100 °C are reached during wort boiling. In a study conducted by Gati et al. [224], in Ghana, 48 bottles of lager beers, comprising 8 different batches from each of the six brands, were analysed. The highest compound detected in the lager beers was TCM with a level of 20 μg L^−1^ and the lowest detected was DBCM with a concentration of 0.3 μg L^−1^. The levels detected were below the WHO guideline values for total daily intake of 300 μg L^−1^, 60 μg L^−1^, 100 μg L^−1^, and 100 μg L^−1^ for TCM, BDCM, DBCM, and TBM in drinking water. However, the health risk may be increased by regular consumption of beers with higher concentrations of THMs, such as those found in some of the samples. In 107 beers (consisting of 27 Chinese brands), THMs were measured by HS-GC with mean and maximum concentrations of 1.2 µg L^−1^ and 5.2 µg L^−1^, respectively. TTHMs were also measured in water samples from different brewery sites. Concentrations ranged from 3.0 to 47 µg L^−1^, with the exception of one sample containing 79 µg L^−1^ [225].

## 4. Conclusions and Recommendations

There were well over 120 different recognised beer styles in 2019, according to the Brewers Association. Each type of beer is unique in its grain bill, maturation, fermenting method, additives, microbiology, and other differences that affect the chemical composition of the finished product. With the increasing number of microbreweries producing craft beer, the development of new technologies, and the use of unusual ingredients in the brewing process, regular checks on the health and safety of beer must be carried out to make sure that beer continues to meet food safety requirements. Craft beer differs from industrial beer in terms of brewing ingredients, fermentation time, processing, and storage, among other things. As most craft beers are not filtered and sterilised, they are not very shelf stable and have a short shelf life, unlike industrial beers, which are filtered and pasteurised and have a longer shelf life. Industrial production is usually a fully automated process, which distinguishes it from production in craft microbreweries. To avoid pollution, especially by heavy metals, mycotoxins, and pesticides, better control of brewing malts for craft beers should be introduced. Systematic monitoring of beer pollutants can help breweries (craft and industrial) and the brewing industry protect themselves from dangerous risks to human and animal health. Therefore, breweries must implement HACCP (Hazard Analysis and Critical Control Points) to control critical points and address risk through prevention.

Many studies have shown the presence of some chemical pollutants in beer, the most popular beverage in the world after water and tea, which can come from different sources. In some cases, they are present in the raw materials (barley and other cereals, hops, and water) as MTs or PCs generated by moulds or phytosanitary treatments, or as environmental contaminants, as in the case of HMs, CAS, PCBs, PCDDS, and PCDFs. Another source of HMs, PTs, and BPs in beer is the brewing equipment (vessels, tanks, pipes, and filters) or containers used for transport or storage of the final product. The results indicate that canned beer contains higher levels of some contaminants, such as TPTs and BPs, due to migration from can coatings compared to glass bottled beer, which appears to be the safest from a health perspective as it contains the lowest levels of these compounds. Chemical pollutants can also be generated during water sterilisation (chlorination), as in the case of THMs and other disinfection by-products. Finally, they can be generated as unintentional by-products during the various stages of beer brewing, such as AA, Bas, or NAs, mainly due to high temperatures and wort composition. However, their residual levels are usually below the maximum levels allowed by international regulations. Steeping, kilning, mashing, boiling, fermenting, and clarifying can inhibit levels of many pollutants. During these stages, many of them are either removed by draff, spent grains, and fermentation residues, diluted or destroyed by thermal treatment, or adsorbed on the surface of solid particles.

Efforts to improve the production, harvesting, and storage of grain, malt, and by-products to reduce the development of moulds, pests, and diseases should be encouraged. Appropriate storage conditions for grain, malting, and brewing by-products are clearly important, such as low water activity and temperature and shorter storage times. In addition, the technological requirements of the brewing process must be strictly controlled, as must the packaging materials for the beer.

## Figures and Tables

**Figure 1 foods-13-01709-f001:**
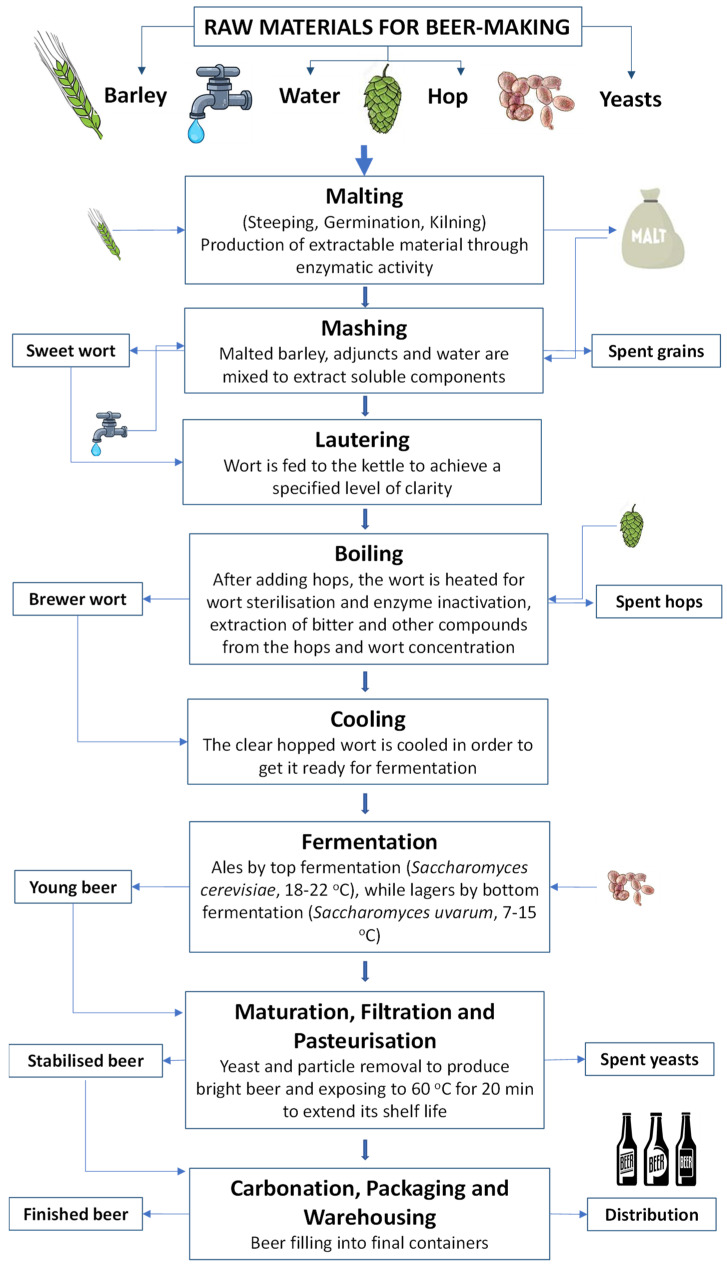
Main stages of the beer-making process.

**Figure 2 foods-13-01709-f002:**
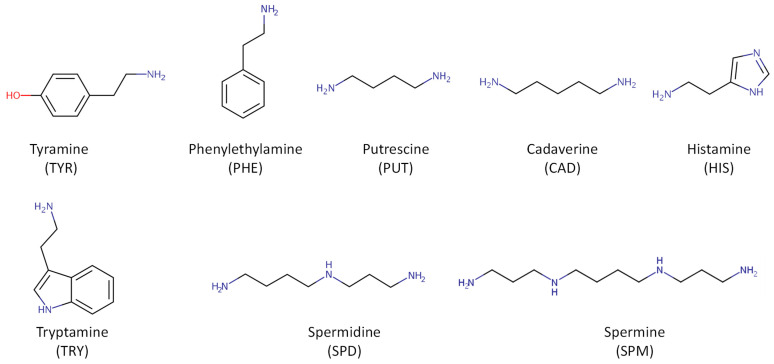
Structural formulas of biogenic amines (BAs).

**Figure 3 foods-13-01709-f003:**
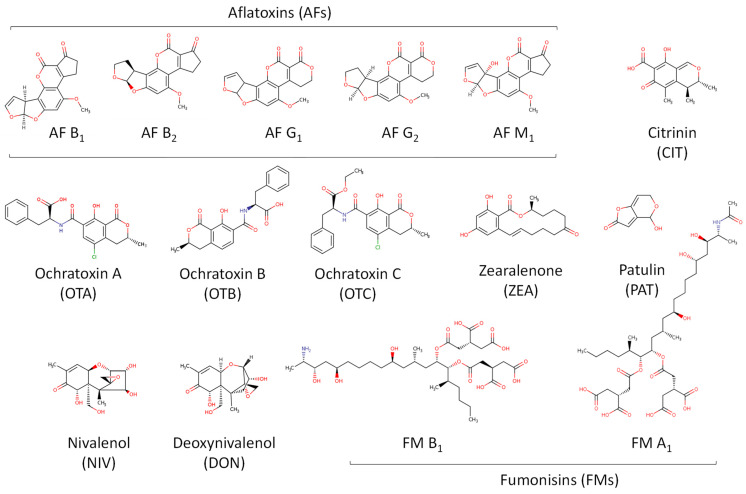
Structures of the main mycotoxins (MTs).

**Figure 4 foods-13-01709-f004:**
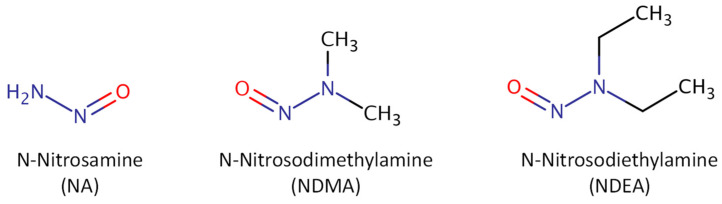
Chemical structures of nitrosamines (NAs).

**Figure 5 foods-13-01709-f005:**
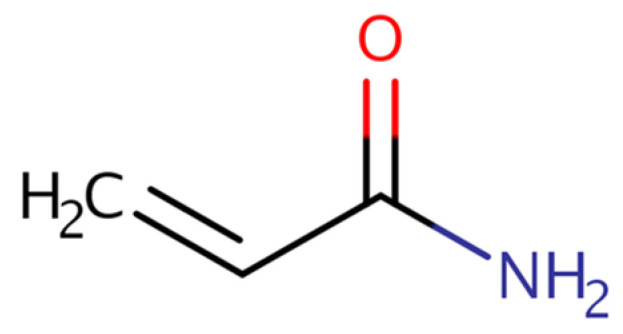
Structure of acrylamide (AA).

**Figure 6 foods-13-01709-f006:**
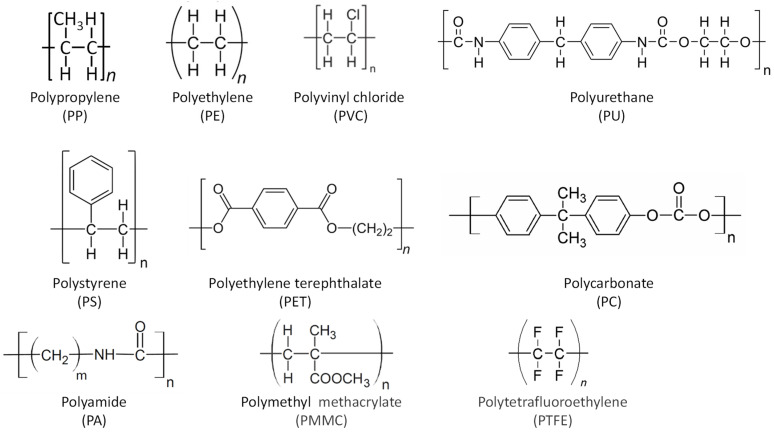
Structures of microplastics (MPs).

**Figure 7 foods-13-01709-f007:**
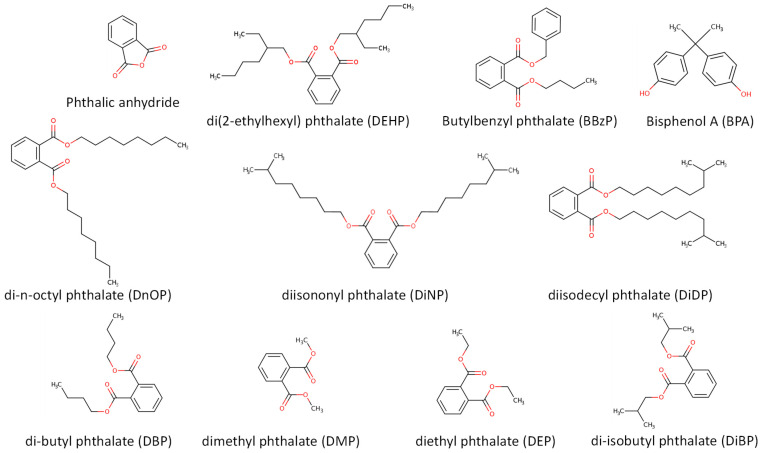
Structures of phthalates (PTs) and bisphenol A (BPA).

**Figure 8 foods-13-01709-f008:**
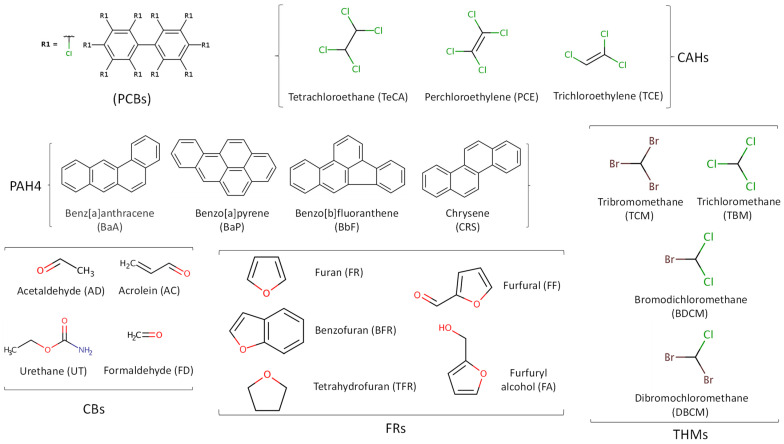
Structures of some of the main representatives of polychlorinated biphenyls (PCBs), chlorinated aliphatic hydrocarbons (CAHs), polycyclic aromatic hydrocarbons (PAHs), carbonyls (CBs), furans (FRs), and trihalomethanes (THMs).

**Table 1 foods-13-01709-t001:** Main chemical pollutants detected during beer making.

Pollutant	Pollution Source	Toxicological Remarks *
Acrylamide (α, β-unsaturated (conjugated) reactive molecule (C_3_H_5_NO)).	Thermal processing.	A range of adverse health effects, including mutagenicity, genotoxicity, carcinogenicity, neurotoxicity, and reproductive toxicity.
Aliphatic chlorinated hydrocarbons (Chlorinated derivatives of non-cyclic hydrocarbons).	Ground or surface water. Technological treatment of drinking water in breweries.	They are dangerous because of their persistence, toxicity, and ability to accumulate in biological systems. They are stored in fat tissue in the human body and cause carcinogenic diseases with prolonged exposure.
Biogenic amines (Organic nitrogen compounds formed by decarboxylation of free amino acids).	Microbial contamination in the brewery. Decarboxylation of free amino acids.	They have a toxic effect on the human body above-limit concentrations.
Bisphenols (Group of chemical compounds related to diphenylmethane based on two hydroxyphenyl functional groups linked by a methylene bridge, with the exception of bisphenol S, P, and M).	Migration from plastic contact materials to raw material and beer.	Bisphenol A is a xenoestrogen, which has hormone-like properties that mimic the effects of oestrogen in the body.
Carbonyls (Carbonyl compounds (carbonyls), mainly including aldehydes and ketones, are a crucial class of oxygen-containing volatile organic compounds (VOCs) in the troposphere).	Atmospheric carbonyls come from both primary emissions (vehicle exhaust, solvent volatilization, industry, and plants) and secondary production (carbonyls generated by the photooxidation of VOCs) from anthropogenic and natural sources.	Some carbonyls harm human health due to their potential mutagenic and carcinogenic properties.
Furans and derivatives (Furan is a 5-membered heterocyclic, oxygen-containing, unsaturated ring compound. Compounds containing the furan ring (as well as the tetrahydrofuran ring) are usually referred to as furans).	They are low-molecular-weight compounds with high volatility found in heat-treated commercial foods and produced through thermal degradation of natural food precursors such as ascorbic acid, amino acids, carbohydrates, unsaturated fatty acids, and carotenoids.	Several of these compounds cause necrosis of target cells within certain organs, including the liver, the kidneys, and the lungs.
Heavy metals (Metal of relatively high densit, or of high relative atomic weight).	Barley, hop, and water. Brewing equipment (pipes, tanks, containers, and filtration equipment) or containers for transporting or storing the final product.	As, Pb, Cd, Cr, Hg, and others. Possibly carcinogenic and accumulates in the human body. Cancer of the skin, lungs, liver, prostate, bones, and bladder. Kidney and liver dysfunction, high blood pressure, liver damage, and bone fragility. Neurotoxicity, respiratory and digestive effects, and neurodegenerative diseases such as Alzheimer’s disease.
Microplastics (Extremely small pieces (<5 mm) of plastic debris),	Disposal and breakdown of consumer products and industrial waste.	The nature of the human health effects and the ultimate damage cannot be predicted at this time.
Mycotoxins (Toxins of natural origin produced as secondary metabolites by microscopic filamentous fungi).	Cultivation of cereals in the field, as well as during storage.	Chemically and thermally very stable compounds. The adverse health effects of mycotoxins range from acute poisoning to long-term effects such as weakening of the immune system and cancer.
Nitrosamines and ATNC (sum of all N-nitroso compounds) (Substances belonging to a group of N-nitroso compounds, i.e., substances that have a covalently bonded nitroso group (NO) to a nitrogen atom in their molecule).	Bacterial contamination. Product of reaction of amines naturally found in barley with nitrogen oxides from drying air, or they can also be transformed from pesticides.	N-nitrosamines are highly toxic, including carcinogenic, mutagenic, embryopathic, and teratogenic effects.
Pesticides (Crop protection products for a wide range of diseases, pests, and weeds, as well as plant growth regulators, including insecticides, fungicides, and others.)	Barley, hops, water, and soil.	They can cause many acute and chronic diseases, such as endocrine disruption, infertility, and abnormal foetal development. Pesticides can also affect the development of the nervous system, leading to problems with coordination, behavioural problems, or delayed physical development. Some pesticides have a negative effect on the immune system and cause allergies. Others are proven carcinogens and teratogens.
Phthalates (Esters of phthalic acids and plasticizers).	Raw materials, but they can also be released from plastic materials that are in a direct contact with beer or intermediates.	They have been shown to be carcinogenic, affect the endocrine system, and can cause premature birth or asthma.
Polychlorinated biphenyls (PCBs) (A mixture of biphenyl molecules substituted with chlorine atoms).	Soil, air, and water. Used widely in electrical equipment like capacitors and transformers.	High chemical, thermal, and biological stability PCBs are very harmful substances that cause liver damage, affect endocrine function, and cognitive function, are carcinogenic and immunotoxic, and cause reproductive and developmental problems.
Polycyclic aromatic hydrocarbons PAHs) (Compounds composed of two or more condensed benzene rings in different configurations with different substituents).	Soil, air, and water. Volcanic eruptions and forest fires.	Smoking, grilling, and roasting increase the levels of PAHs in drinks. PAHs have been shown to have mutagenic and carcinogenic effects, but these effects and their severity depend on the chemical structure.
Trihalomethanes (Compounds with single-carbon substituted halogens (CHX_3_), where X can be fluorine, chlorine, bromine, or iodine, or a group of these).	Source of water (municipal, surface or groundwater used in breweries) and system used for water sterilisation.	Exposure to higher amounts of trihalomethanes may cause reproductive problems and birth defects with DNA damage.

* Agency for Toxic Substances and Disease Registry (ATSDR), U.S. Department of Health and Human Services. https://www.atsdr.cdc.gov/ (accessed on 13 January 2024).

**Table 2 foods-13-01709-t002:** MTs detected from 2014 to 2018 in monitored beer samples [88].

Year	MTs	Samples (No)	+Samples (No)	+Samples (%)	Levels *	Below Limit **
2014	AFs	35	0	0	-	
	DON	43	28	65	2.33–22.5	√
	ZEA	27	0	0	-	
	T-2 and HT-2	24	8	33	0.17–0.71	√
	OTA	47	33	70	1.4–141	√
2015	AFs	37	0	0	-	
	DON	47	14	30	2.01–29.3	√
	ZEA	35	0	0	-	
	T-2 and HT-2	35	18	51	0.04–0.9	√
	OTA	50	35	70	1.8–28.8	√
2016	AFs	38	0	0	-	
	DON	73	24	33	2.12–10.7	√
	ZEA	25	1	4	0.41	
	T-2 and HT-2	25	4	16	0.04–0.82	√
	OTA	78	62	80	1.0–134	√
2017	AFs	35	0	0	-	
	DON	50	29	58	2.09–13.9	√
	ZEA	3	0	0	-	
	T-2 and HT-2	29	7	24	0.3–0.85	√
	OTA	4955	45	92	1.3–77.3	√
2018	AFs	57	0	0	-	
	DON	40	44	77	2.03–17.0	√
	ZEA	43	0	0	-	
	T-2 and HT-2	67	38	88	0.05–1.8	√
	OTA		51	76	1.4–56.1	

* ng L^−1^ for AFs and OTA and μg L^−1^ for DON ZEA, and T-2 and HT-2 mycotoxins. ** The above limit has been derived from the highest concentration found of a given mycotoxin, its tolerable weekly intake (TWI), or tolerable daily intake (TDI) and the recommended consumption of beer under moderate drinking conditions, which is two 0.5 L beer for men and two 0.3 L beer for women [4].

**Table 3 foods-13-01709-t003:** Quantities (%) of certain pesticide residues remaining during the malting process.

Pesticides	Log *K*_OW_	Stage			References
		Steeping	Germination	Kilning	
Cyproconazole	3.1	47	38	31	[114]
Diniconazole	4.3	70	61	39	[114]
Epoxiconazole	3.4	62	53	38	[114]
Ethiofencarb	2.0	3	1	5	[112]
Fenitrothion	3.4	52	31	13	[113]
Flutriafol	2.3	43	35	30	[113]
Malathion	2.8	45	20	14	[113]
Mepronil	3.8	24	6	30	[112]
Myclobutanil	2.9	59	42	36	[113]
Nuarimol	3.2	64	57	51	[113]
Pendimethalin	5.2	85	67	49	[113]
Phentoate	3.7	27	4	18	[112]
Propiconazole	3.6	50 55	10 43	55 30	[112,113]
Tebuconazole	3.7	56	45	37	[114]
Triadimefon	3.1	24	5	30	[112]
Triadimenol	3.1	36	13	47	[112]
Triflumizole	4.4	38	11	9	[112]
Trifluralin	5.3	80	65	50	[113]

## Data Availability

No new data were created or analyzed in this study. Data sharing is not applicable to this article.

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
