# Peer review of "Understanding How Chemical Pollutants Arise and Evolve in the Brewing Supply Chain: A Scoping Review"

_foods, 2024, doi:10.3390/foods13111709_

Round 1

Reviewer 1 Report

Comments and Suggestions for Authors

Manuscript ID: foods-3025060

Title: Understanding how Chemical Pollutants Arise and Evolve in the Beer Brewing Process: A Scoping Review

The manuscript is interesting, but I would suggest to strengthen the analysis of chemical pollulants in malting. Furthermore, I would make a further effort to add the legal limits for pollutants established by mandatory legislation around the world. Given the tendency to reuse spent grain in the food industry, is there any research relating to their content in chemical pollulants after drying and grinding? The authors cited some chemical pollulants in craft beers but it should be interesting if they highlighted the difference between industrial and craft beers in terms of possible contaminations. What about organic beers? Authors collected a great numbers of data; however, when reading the manuscript, there is a sort of saturation effect. It would be helpful to give the manuscript a more linear structure

In addition to responses to the above-mentioned considerations, I ask to make the following changes.

Title: please, change “Beer Brewing Process” to “brewing supply chain”

Keywords: please, change “beer-making” to “brewing”

Introduction

Line 35: please, remove “often”

Lines 53-56: The stage (v) occurs after boiling… please, correct

Lines 58-60: not only ethanol and carbon dioxide but also other compounds……

Lines 68-69: please, remove “This is one of many reasons 68 why beer is perceived as a healthy drink.”

Lines 91-93: “It is therefore essential that breweries have a hazard analysis and critical control 91 points (HACCP) programme in place, a preventative, methodological approach to the 92 safety of beer that controls critical points and addresses risk through prevention.”…. please, remove…. It is well known that for about 30 years European food companies have been obliged to apply HACCP

3.3 Mycotoxins

Line 290 and throughout the manuscript: use italics for the names of genus and specie

Author Response

First of all, thank you for your kind and useful comments.

The manuscript is interesting, but I would suggest to strengthen the analysis of chemical pollulants in malting. Furthermore, I would make a further effort to add the legal limits for pollutants established by mandatory legislation around the world. Given the tendency to reuse spent grain in the food industry, is there any research relating to their content in chemical pollulants after drying and grinding? The authors cited some chemical pollulants in craft beers but it should be interesting if they highlighted the difference between industrial and craft beers in terms of possible contaminations. What about organic beers? Authors collected a great numbers of data; however, when reading the manuscript, there is a sort of saturation effect. It would be helpful to give the manuscript a more linear structure

  • We believe that there are numerous references to chemical contaminants during malting, particularly pesticides. For certain pollutants, the maximum levels allowed by some legislations have been included. However, it is very difficult to include these values for all the pollutants studied, as there are many existing laws and not all countries have set maximum limits. Brewer’s spent grain (BSG) is a byproduct of the brewing process that accounts for approximately 85% of the waste produced by a brewery. BSG are mainly used in agriculture as a soil amendment to fertilize crops or feedstuffs for aquaculture and to produce biogas. However, the most popular option for brewers is to use them to supplement livestock feed. If BSG is being used in human food (replacing flour in baked goods such as snack bars, bread, crackers and pet food), the FDA mandates that you have a Hazard Analysis and Critical Control Points (HACCP) food safety plan. However, research relating to their content in chemical pollutants after drying and grinding is scarce. The main difference between industrial and craft beers in terms of possible pollution is related to the control carried out during the brewing process and the existence of an HACCP process. Regarding organic beers, the main difference is the presence of pesticide residues.

In addition to responses to the above-mentioned considerations, I ask to make the following changes.

Title: please, change “Beer Brewing Process” to “brewing supply chain”

  • According to your suggestion, the title has been changed.

Keywords: please, change “beer-making” to “brewing”

  • It has been changed

Introduction

Line 35: please, remove “often”

Lines 53-56: The stage (v) occurs after boiling… please, correct

Lines 58-60: not only ethanol and carbon dioxide but also other compounds……

Lines 68-69: please, remove “This is one of many reasons 68 why beer is perceived as a healthy drink.”

Lines 91-93: “It is therefore essential that breweries have a hazard analysis and critical control 91 points (HACCP) programme in place, a preventative, methodological approach to the 92 safety of beer that controls critical points and addresses risk through prevention.”…. please, remove…. It is well known that for about 30 years European food companies have been obliged to apply HACCP

3.3 Mycotoxins

Line 290 and throughout the manuscript: use italics for the names of genus and specie

  • All suggestions have been incorporated in the revised version.

Reviewer 2 Report

Comments and Suggestions for Authors

This reviewer believes the manuscript addresses an interesting and important topic. However, I recommend revising some details.

1. In general, the text is well-written, but some confusing sentences need to be reformulated for better comprehension. Also, the whole text would benefit from another in-depth language check to eliminate grammar and confusing language errors

2.  In lines 63-69 authors say:

 "Nowadays, the benefits of reasonable beer consumption for human health are increasingly emphasised due to the absence of negative properties and the presence of positive ones, such as significant amounts of minerals, antioxidants, vitamins and other healthy substances, as well as low sugar content [4]. Many studies show that moderate drinkers have lower death rates from all causes, but particularly from cardiovascular disease, than heavy drinkers and those who don't drink at all. This is one of many reasons why beer is perceived as a healthy drink".

Although the information is well-referenced, I recommend not emphasizing beer as a healthy beverage or the habit of drinking beer as a path to prevent disease since this is irrelevant to the study.

Also, the very same referenced article has the following conclusion:" Patients with increased risk for specific diseases, for example women with familiar history of breast cancer, or subjects with familiar history of early CVD or cardiovascular patients should discuss their drinking habits with their physician. No abstainer should be advised to drink for health reasons."

3.   The abstract should include the aim of the work.

4.   The main aspects of each class of chemical pollutants are well described. However, it would be very interesting to include a table summarising the maximum levels that international regulations allow and comparing it with the level found in beers. An

5.   "Are there no other terms to refer to brewing other than 'brewing' and 'beer-making'?

6.  I recommend including a section to discuss the pathways that, from the authors' perspective, should be done to control or overcome the problems related to the chemical pollutants.  

7. The authors comment on the main problems of each pollutant in human health, but is there any information about their combined effect?

The mentioned issues and other additional comments/suggestions are highlighted in the attached file. I hope the comments are constructive and help the authors to improve the manuscript.

Comments on the Quality of English LanguageThe whole manuscript text would benefit from another in-depth language check to eliminate grammar and confusing language errors

Author Response

First of all, thank you for your kind and useful comments.

This reviewer believes the manuscript addresses an interesting and important topic. However, I recommend revising some details.

1.In general, the text is well-written, but some confusing sentences need to be reformulated for better comprehension. Also, the whole text would benefit from another in-depth language check to eliminate grammar and confusing language errors

2.In lines 63-69 authors say:

"Nowadays, the benefits of reasonable beer consumption for human health are increasingly emphasised due to the absence of negative properties and the presence of positive ones, such as significant amounts of minerals, antioxidants, vitamins and other healthy substances, as well as low sugar content [4]. Many studies show that moderate drinkers have lower death rates from all causes, but particularly from cardiovascular disease, than heavy drinkers and those who don't drink at all. This is one of many reasons why beer is perceived as a healthy drink".

Although the information is well-referenced, I recommend not emphasizing beer as a healthy beverage or the habit of drinking beer as a path to prevent disease since this is irrelevant to the study.

  • The term “healthy beverage” has been deleted.

Also, the very same referenced article has the following conclusion:"Patients with increased risk for specific diseases, for example women with familiar history of breast cancer, or subjects with familiar history of early CVD or cardiovascular patients should discuss their drinking habits with their physician. No abstainer should be advised to drink for health reasons."

3.The abstract should include the aim of the work.

- It has been included

4.The main aspects of each class of chemical pollutants are well described. However, it would be very interesting to include a table summarising the maximum levels that international regulations allow and comparing it with the level found in beers

- For certain pollutants, the maximum levels allowed by some legislations have been included. However, it is very difficult to include these values for all the pollutants studied, as there are many existing laws and not all countries have set maximum limits.

5.Are there no other terms to refer to brewing other than 'brewing' and 'beer-making'?

- Both are the terms more usually used.

6.I recommend including a section to discuss the pathways that, from the authors' perspective, should be done to control or overcome the problems related to the chemical pollutants.  

- This comment is included in Conclusions.

7.The authors comment on the main problems of each pollutant in human health, but is there any information about their combined effect?

- There is no data because it is impossible to carry out a global evaluation.

The mentioned issues and other additional comments/suggestions are highlighted in the attached file. I hope the comments are constructive and help the authors to improve the manuscript.

- More than 95% of your additional comments, which are highlighted in the attachment, have been considered in the revised version.

Round 2

Reviewer 1 Report

Comments and Suggestions for Authors

Concerning my comments:

The manuscript is interesting, but I would suggest to strengthen the analysis of chemical pollulants in malting. Furthermore, I would make a further effort to add the legal limits for pollutants established by mandatory legislation around the world. Given the tendency to reuse spent grain in the food industry, is there any research relating to their content in chemical pollulants after drying and grinding? The authors cited some chemical pollulants in craft beers but it should be interesting if they highlighted the difference between industrial and craft beers in terms of possible contaminations. What about organic beers? Authors collected a great numbers of data; however, when reading the manuscript, there is a sort of saturation effect. It would be helpful to give the manuscript a more linear structure

the authors should have edited the manuscript accordingly and not just given me concise and sometimes incorrect answers (see for example, the differences between craft and industrial beer contaminants.

Furthermore, the authors should re-organize the manuscript in a more linear manner. Currently, it is a large mass of poorly organized information.

Author Response

Response to reviewer 1 (round 2)

the authors should have edited the manuscript accordingly and not just given me concise and sometimes incorrect answers (see for example, the differences between craft and industrial beer contaminants.

  • On pages 8, 9, 23 and 25 you can explore some differences between the contaminants found in craft beer and industrial beer.

Furthermore, the authors should re-organize the manuscript in a more linear manner. Currently, it is a large mass of poorly organized information.

  • It is true that the manuscript contains a lot of information about pollutants during brewing. However, we do not understand your statement: "the authors should re-organise the manuscript in a more linear way.". Poorly organized information? Honestly, we think that the manuscript is well organised. It includes a general introduction to the topic, the methodology, followed by the description of the main (majority and minority) contaminants found during beer production in the subsections (3.1 to 3.8). All this information is clearly described in the different subsections of the manuscript. There is also a section on conclusions.